# MatMuls are Enough for Efficient and Performant Linear-Time Attention

## Abstract

Transformers, despite empowering current AI revolution, are bottlenecked by sub-optimal hardware utilization and quadratic runtime complexity of softmax attention w.r.t. input sequence length. Many recent architectures aspire to bring the complexity down to sub-quadratic level without compromising modeling quality. However, they are either much slower on all but very long sequences or rely on low-level code tailored to a narrow subset of modern hardware. To simultaneously achieve linear complexity, hardware efficiency, and portability, we completely eliminate softmax from self-attention; and remove, reduce, modify, or rearrange other transformations in the Transformer block. The resulting architecture, DenseAttention Network, is composed entirely of dense matrix multiplications in the attention which allows for efficient training and inference in both quadratic and linear modes. It outperforms standard Transformer and its counterparts in language modeling benchmarks and surpasses previous Transformer-based SOTA by 5% on challenging Long Range Arena. DenseAttention model written in plain PyTorch is up to 22% faster even on small context sizes, and by orders of magnitude on longer sequences, than Transformer with low-level FlashAttention kernel.

## 1 Introduction

Since its inception almost a decade ago, the softmax self-attention mechanism (Bahdanau et al., 2015; Graves, 2014) and Transformer block (Vaswani et al., 2017), built upon it, have become a dominant deep learning paradigm. Transformer is the foundation of multiple state-of-the-art architectures in diverse modalities, such as natural language processing (Devlin et al., 2019; Raffel et al., 2019), vision (Dosovitskiy et al., 2021; Kirillov et al., 2023), speech recognition (Radford et al., 2022), and even tabular data (Arik and Pfister, 2019). Notably, it's the core component of Large Language (Touvron et al., 2023a; Brown et al., 2020) and Multi-Modal (Dubey et al., 2024; Yang et al., 2025) Models.

Yet Transformer architecture is bottlenecked by $O(N^2)$ time and space complexity of self-attention w.r.t. context length $N$ which substantially impedes its ability to process long sequences by rendering its computation slow and expensive. Partially, this limitation is alleviated by FlashAttention series of low-level, hardware-efficient kernels (Dao et al., 2022b; Dao, 2024; Shah et al., 2024) designed to optimize and streamline computation of self-attention on a subset of modern GPUs. FlashAttention reduces memory consumption to $O(N)$ but does not circumvent the quadratic runtime complexity of attention.

Another major limitation inherent to Transformer is the computational inefficiency of the constituents which make the architecture work seamlessly. As reported by Ivanov et al. (2021), matrix multiplications account for 99.8% of total FLOPs during BERT pre-training and only 61% of runtime, the discrepancy being caused by low arithmetic intensity of memory bound operations, namely, LayerNorm, softmax and other activations as well as elementwise operations (see Appendices C, D for the underlying reasoning and possible mitigation measures).

Numerous extensions and modifications to the standard Transformer (Katharopoulos et al., 2020; Beltagy et al., 2020; Choromanski et al., 2022; Hua et al., 2022; Kacham et al., 2024) have been proposed in order to mitigate the restrictive $O(N^2)$ complexity (extended discussion of related work is available in Appendix A). However, as these architectures in general rely on non-linear, memory-intensive and sparse operations to a much greater degree than traditional attention mechanism, their

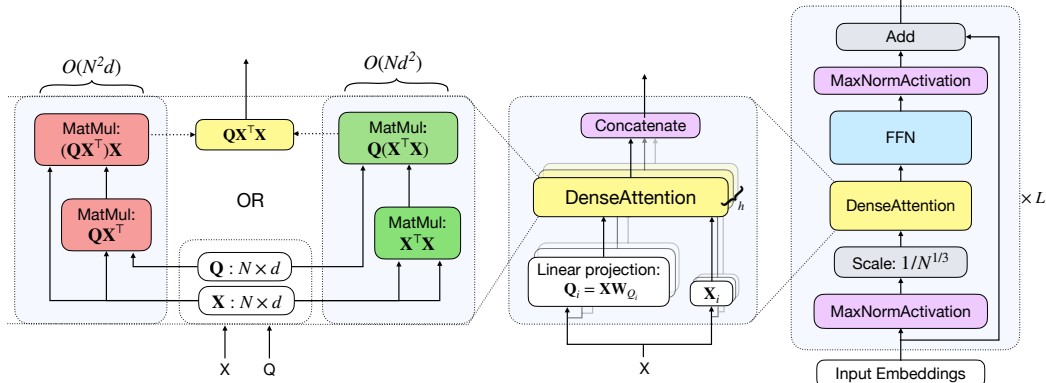

Figure 1: DANet architecture. **Left:** DenseAttention mechanism. **Center:** multi-head interpretation. **Right:** the entire DenseAttention Network. DenseAttention, the core component of the arhitecture, is composed entirely of MatMuls.

throughput in terms of tokens per second and hardware utilization are subpar in comparison with the latter on all but large sequence lengths (Tay et al., 2022; Dao et al., 2022b). Besides, some report (Xiong et al., 2022; Tay et al., 2023; Buckman and Gelada, 2024), that their modeling capabilities may be limited in comparison with full-rank exact attention while their conceptual complexity and incompatibility with standard architectures prevents their widespread adoption. Recently, SSM/RNN-inspired linear-time architectures (Yang et al., 2024; Gu and Dao, 2024; Beck et al., 2024) have been shown to perform on par with Transformer, but they also rely on specialized CUDA/Triton (Tillet et al., 2019) kernels, limiting their use to a subset of modern GPUs.

Thus, we aim to achieve three main goals: (i) To create ***hardware efficient yet hardware-agnostic architecture***, having the arithmetic intensity ratio as high as possible and compatible with wide range of devices. (ii) To create an algorithm which would efficiently process both long and short sequences, preferably with $O(N)$ ***time and space complexity*** (iii) To make the resulting ***architecture as simple as possible***, closely resembling original Transformer, so it can serve as a drop-in replacement for the latter and be easily adopted by both research and practitioners communities.

We accomplished all of these goals with DenseAttention and DenseAttention Network (DANet) blocks. This architecture is a straight-forward simplification of the traditional Transformer architecture characterized by no additional complexities and by *elimination* of almost all computationally inefficient (as detailed in App. C, D) elements of the original architecture: biases in all layers, masks, dropout, one of residual connections, and all attention projection matrices except $W_Q$. Additionally, we reduce number of heads and propose a Local–ShiftedLocal–Global attention layers scheme to boost interactions among nearby tokens in extremely long sequences. And most importantly, we remove Softmax inside self-attention. It results in the whole scaled dot-product attention mechanism becoming just a composition of matrix multiplications, which can be done in any order by associative property of matrix multiplication. This duality allows to calculate DenseAttention using either $O(N^2d)$ or $O(Nd^2)$ FLOPs (Fig. 1), the second option having linear time and space complexity.

We replace LayerNorms with a novel MaxNormActivation, which scales token representations by their $l_\infty$ norm. This makes DenseAttention numerically stable even in an older reduced-range *fp16* format. Together with a simple implementation, runnable on every PyTorch-supported platform, it facilitates access to efficient and performant linear-time architectures for a wider audience.

DenseAttention does not approximate or substitute softmax with any other selectivity mechanics. This makes it fundamentally different from Linear Attention (Katharopoulos et al., 2020) family of models as it doesn't employ non-negative transforms of queries and keys and reweighing attention scores by their sum. It's neither a State-Space-Model (SSM) or a Linear RNN because it has no decay or gating modules, as in Gu and Dao (2024); Yang et al. (2024), and it natively supports bidirectional context processing (see App. A for extended discussion).

To empirically validate the architecture, we conduct experiments in language modeling by replacing Transformer modules in BERT (Devlin et al., 2019) and Llama-like (Touvron et al., 2023a) models

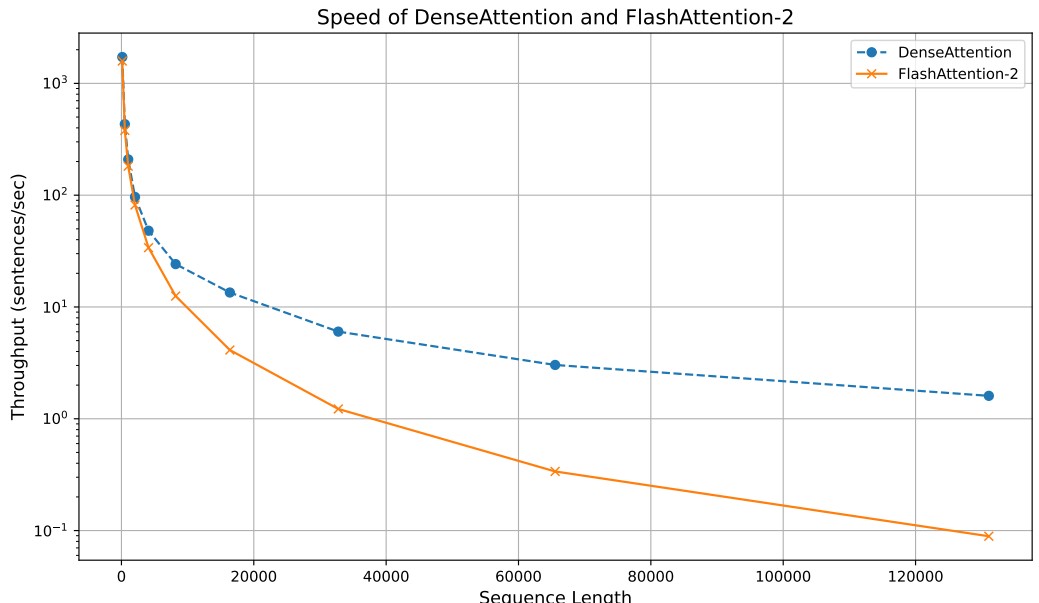

Figure 2: Comparison of inference speed between DenseAttention and FlashAttention-2 (Dao, 2024) based 360M-parameters models across sequence lengths on a NVIDIA A100 40GB. DenseAttention is faster for all sequence lengths despite being written in plain PyTorch with no hardware optimizations. Both models are used with the torch.compile() module.

with DenseAttention Network modules and pre-training them from scratch on sequences up to 16k tokens. The model achieves quality metrics that exceed modern Transformer baselines and exceed other architectures on GLUE (Wang et al., 2018), while enjoying significantly faster training and inference speed both in $O(N^2)$ and $O(N)$ regimes (Fig. 2). We then test it on diverse sequence modeling tasks from challenging Long Range Arena (LRA) benchmark (Tay et al., 2021) and achieve a new SOTA result across all of the transformer-based models, including all of Linear Attention (Katharopoulos et al., 2020) derived ones, and even competing with SSMs (Gu et al., 2022a).

To the best of our knowledge, we are the first to train a competitive BERT-Large sized encoder language model with only matrix multiplications in the attention layer. To facilitate further research, we release the code and the weights of this model at <...>.

## 2 BACKGROUND

Here we give a brief exposition of essential elements of Transformer architecture and their variations.

Standard Transformer block consists of self-attention and feed-forward-network (FFN) sub-blocks (Vaswani et al., 2017). Let $\mathbf{X} \in \mathbb{R}^{N \times d}$, where $N$ is the sequence length and $d$ is an embedding dimension of one token. Define $\mathbf{Q} = \mathbf{X}\mathbf{W}_Q$ as queries, $\mathbf{K} = \mathbf{X}\mathbf{W}_K$ as keys, and $\mathbf{V} = \mathbf{X}\mathbf{W}_V$ as values, where $\mathbf{W}_Q, \mathbf{W}_K, \mathbf{W}_V \in \mathbb{R}^{d \times d_h}$ are learnable parameters. Default implementations in some models (e.g. Devlin et al. (2019); Yang et al. (2025)) also add biases to $\mathbf{Q}, \mathbf{K},$ and $\mathbf{V}$. Then the *Scaled Dot-Product Attention* is formulated as:

$$\text{Attention}\,(\mathbf{X}) = \text{Attention}(\mathbf{Q}, \mathbf{K}, \mathbf{V}) = \text{Softmax}\left(\frac{\mathbf{Q}\mathbf{K}^\top}{\sqrt{d_h}} + \mathbf{M}\right)\mathbf{V}, \qquad (1)$$

with Softmax applied row-wise and mask $\mathbf{M} \in \mathbb{R}^{N \times N}$ with values 0 or $-\infty$ which effectively disables some positions from calculation to account for causal sequence processing or to conceal 'PAD' token used for batch processing of sequences with different lengths.

Essentially, all transformer-based models use some form of Multi-Head Attention which has $H$ heads. Attention equation 1 is calculated for each head independently and the results are concatenated

along the embedding dimension and projected back to full block's output dimension by a matrix $\mathbf{W}_O \in \mathbb{R}^{d \times d_{out}}$:

$$\text{MultiHeadAttn}(\mathbf{Q}, \mathbf{K}, \mathbf{V}) = \text{Concat}(\text{head}_1, \ldots, \text{head}_H)\mathbf{W}_O \qquad (2)$$

Feed-Forward Network which follows self-attention is composed of two linear layers and an activation (usually ReLU or GeLU) in between. Intermediate inner dimension between the two layers is usually chosen to be 4x larger than input/ output dimension. Finally, a LayerNorm layer and a residual connection are applied around both blocks, their relative positions dictated by PreNorm or PostNorm architectural choice (Xiong et al., 2020). The formulation of the whole Transformer layer $l$ with PreNorm is:

$$\mathbf{X}'_l = \mathbf{X}_l + \text{Attention}(\text{LayerNorm}(\mathbf{X}_l))$$
$$\mathbf{X}_{l+1} = \mathbf{X}'_l + \text{FFN}(\text{LayerNorm}(\mathbf{X}'_l))$$

Thus, each full Transformer block has two LayerNorms and two residual connections.

## 3 DESIGNING DENSEATTENTION

In this section, we sketch the design of the DenseAttention architecture step-by-step by motivating each of the changes in the architecture as compared to Transformer. Then we outline an extension designed to efficiently adapt a component widely and successfully used in contemporary models: LocalAttention layers.

### 3.1 DENSEATTENTION

**Biases and masking.** First, we note that if there are no biases in all linear layers throughout the Transformer block, then for a row vector $\mathbf{0}_d^\top = [0, 0, \ldots, 0]_{1 \times d}$ : MultiHeadAttn$(\mathbf{0}_d^\top \mathbf{W}_Q, \mathbf{K}, \mathbf{V}) = \mathbf{0}_d^\top$, FFN$(\mathbf{0}_d^\top) = \mathbf{0}_d^\top$, and LayerNorm$(\mathbf{0}_d^\top) = \mathbf{0}_d^\top$, i.e. zero vector stays intact when acted upon by all components of the Transformer module. So we *refrain from using biases* throughout the new block, fix representation of the "PAD" token at the output of embedding layer to $\mathbf{0}_d^\top$, and *remove padding masking* from the self-attention layer.

**Numerical instabilities.** In contrast, removing Softmax without any substitutions proves to be a challenging task: in its absence, attention outputs become unbounded, which can lead for them to either diverge to $\infty$ or shrink to 0. We formalize this statement with the following proposition considering simplified version of the new mechanism where $\mathbf{W} = \mathbf{W}_Q \mathbf{W}_K^\top$ and $\mathbf{W}_V = \mathbf{I}$:

**Proposition 1.** *Let* $\mathbf{X} \in \mathbb{R}^{N \times d}$ *and* $\mathbf{W} \in \mathbb{R}^{d \times d}$ *be matrices composed of i.i.d. random variables, respectively* $X_{ij}$ *with* $\mathbb{E}[X_{ij}] = 0$, $\text{Var}(X_{ij}) = \sigma_X^2$, *and* $W_{km}$ *with* $\mathbb{E}[W_{km}] = 0$, $\text{Var}(W_{km}) = \sigma_W^2$. *Let* $X_{ij}$ *and* $W_{km}$ *also be independent for all* $i, j, k, m$. *Then each element of the matrix* $\mathbf{Y} = \mathbf{X}\mathbf{W}\mathbf{X}^\top \mathbf{X} \in \mathbb{R}^{N \times d}$ *has zero expectation and variance* $\sigma_Y^2 \geq N d^2 \sigma_X^6 \sigma_W^2$.

Essentially, it means that variance of an output grows at least as a cube of an input variance in the new architecture layer. And since $\sigma_Y^2$ along with tail probability $\mathbb{P}(|Y_{ij}| \geq t)$ are not bounded from above and depend on the form of an unknown distribution, we can't just fix $\sigma_X^2$ to ensure numerical stability e.g., by using LayerNorm. It becomes especially important in case of low-precision formats with reduced dynamic range such as fp16. We confirm it empirically in our ablation study (Table 12) as incorporation of LayerNorm leads to a prompt and unrecoverable numerical instability early on during training.

**MaxNorm.** Instead of using $l_2$ norm, we enforce $\max(|X_{ij}|) \leq a$ for some positive $a$ which is equivalent to setting fixed $l_\infty$ norm for the inputs. Consequently, even in worst case scenario where

$$X_{ij} = a \text{ for } \forall\, i, j \qquad (3)$$

it holds for $\mathbf{Z} = \mathbf{X}\mathbf{X}^\top \mathbf{X} \in \mathbb{R}^{N \times d}$:

$$\max(|Z_{ij}|) \leq N d a^3, \qquad (4)$$

i.e. $l_\infty$ norm of output values is bounded above. Furthermore, we make the following observation:

**Proposition 2.** *If elements $W_{km}$ of $\mathbf{W}$ are i.i.d normal variables with mean 0 and variance $\sigma_W^2$, independent with $\forall\, X_{ij}$, then $\mathrm{Var}[(\mathbf{XW})_{pq}] \leq \sigma_W^2 a^2 d$.*

It follows from **Prop. 2.** that $\sigma_W$ and $a$ can be chosen such that $\mathbb{P}[|(XW)_{pq}| \geq \epsilon] \leq \delta$ for some $\epsilon > 0, \delta > 0$ depending on $\sigma_W$ and $a$. Thus, we can assume that the matrix product $\mathbf{Y} = \mathbf{XWX}^\top\mathbf{X} \in \mathbb{R}^{N \times d}$ will not explode with right selection of priors.

Specifically, we set $a = \frac{1}{N^{\frac{1}{3}}}$, so that equation 4 becomes $\max(|Z_{ij}|) \leq d$. We choose not to downscale inputs by further degree, e.g. by $d\sqrt{n}$ because resulting small values may hurt modeling quality during training in low-precision formats (fp16 and bf16).

We fix each embedding vector $\mathbf{X}_i$ to have constant $l_\infty$ norm of 1 by applying our novel *MaxNormActivation* or, more succinctly, *MaxNorm* function:

$$\mathrm{MaxNormActivation}(\mathbf{X}_i) = \frac{\mathbf{X}_i}{\max_j(|\mathbf{X}|_{ij}) + \epsilon}$$

where $\epsilon$ is a very small number put to prevent division by 0. Note that similarly to *RMSNorm* (Zhang and Sennrich, 2019), MaxNormActivation doesn't center its inputs. However, it uses $l_\infty$ norm instead of $l_2$ and doesn't have learnable *scale* and *bias* parameters as in Zhang and Sennrich (2019); Ba et al. (2016) by default. When applying MaxNorm *before* the attention block, we scale outputs by a constant factor of $\frac{1}{N^{\frac{1}{3}}}$.

While in theory $l_2$ and $l_\infty$ are bounded by a constant factor of each other, which is implied by notion of norm equivalence, $l_2$ norm still doesn't prevent attention values from vanishing or exploding, as further analyzed and exemplified in Appendix B.5.

**Linear complexity.** Consequently, it allows the *removal of Softmax*. This not only lifts a major computational and memory inefficiency, which could be alleviated with clever low-level algorithms as in Dao et al. (2022b); Rabe and Staats (2021), but also transforms the attention mechanism into a raw product of three matrices $\mathbf{QK}^\top\mathbf{V}$. Exploiting associative property of matrix multiplication, we can compute the product as either $(\mathbf{QK}^\top)\mathbf{V}$ which yields $2N^2d$ FMA operations, or $\mathbf{Q}(\mathbf{K}^\top\mathbf{V})$ which yields $2Nd^2$ FMA operations and is linear w.r.t $N$ both in time and memory complexity. For causal language modeling, attention can also be computed efficiently in linear time by applying a well-known *chunk-wise parallel* algorithm, described in e.g, Sun et al. (2024); Gu and Dao (2024).

We can utilize both methods interchangeably depending on what's more favorable given particular values of $N$ and $d$. $O(N)$ complexity gives way to processing very large sequences in linear time with the same result as if done in traditional $O(N^2)$ paradigm as it calculates exactly the same all $N \times N$ pairwise interactions but just in another order.

**Further reductions.** Next, we consider *reducing number of heads* in the multi-head attention to increase their arithmetic intensity (see Appendix D). As extensive research efforts have shown (Bhojanapalli et al., 2020; Voita et al., 2019; Kovaleva et al., 2019; Michel et al., 2019), significant portion of heads in multi-head attention are redundant, output low-rank representations and can be pruned without decrease in quality in downstream tasks, at least for BERT-sized models. Specifically, Bhojanapalli et al. (2020) find that increasing number of heads past a certain threshold degrades performance in BERT. Motivated by this, we propose increasing $d_h$ from conventional value 128 up to 1024. In case of BERT example from App. D it leads to a single-head attention.

We note that the matrix $\mathbf{W} = \mathbf{W}_Q\mathbf{W}_K^\top$ in the expression $\mathbf{QK}^\top = \mathbf{XW}_Q\mathbf{W}_K^\top\mathbf{X}^\top$ is essentially low-rank as in standard attention $d_h \ll d$. But in our implementation this rank is much higher, in the extreme case being equal to $d$. It results in multiplication of two high or full rank matrices. That is a redundant operation from DL perspective because composition of linear maps is just another linear map which could be learned using half of the parameters. Thus, we decide to keep the $\mathbf{W}_Q$ and discard $\mathbf{W}_K$, effectively *merging the two matrices*.

We also decide to *remove LayerNorm and residual connection* between attention and FFN sub-blocks as it improves computational efficiency of the architecture and appears not to hinder model performance. This leads to yet another simplification in the model design: $\mathbf{W}_V$ and $\mathbf{W}_O$ also *become redundant and get eliminated* by similar reasoning as in the case of $\mathbf{W}_Q$, because there are no more non-linearities between attention outputs and FFN block.

**DANet.** Following the reductions and eliminations, the new attention mechanism in the case of a single head is formulated as:

$$\text{DenseAttn}\left(\mathbf{X}\right) = \mathbf{X}\mathbf{W}_Q\mathbf{X}^\top\mathbf{X} \in \mathbb{R}^{N \times d}$$

And in the case of multiple heads $H$, $\mathbf{X}$ in places of keys and values is simply sliced into $H$ chunks along the embedding dimension, with $\mathbf{X}_h \in \mathbb{R}^{N \times d_h}$:

$$\text{DenseAttn}_h\left(\mathbf{X}\right) = \mathbf{X}\mathbf{W}_{Q_h}\mathbf{X}_h^\top\mathbf{X}_h \in \mathbb{R}^{N \times d_h}$$
$$\text{DenseAttn}\left(\mathbf{X}\right) = \text{Concat}_h[\text{DenseAttn}_h\left(\mathbf{X}\right)]$$

We call our attention algorithm "DenseAttention" and the entire block as "DenseAttention Network" or DANet (spelled "dah-net") because it basically consists of dense matrix multiplications with little else. We notice that DenseAttention in multi-head setting resembles popular multi-query attention design from Shazeer (2019) as it also calculates different representations only for Queries.

To complete the DenseAttention Network, we apply MaxNormActivation and residual connection to outputs of FFN. Final architecture to the layer $l$ can be summarized as follows:

$$\mathbf{X}'_l = \text{DenseAttn}(\text{MaxNormActivation}(\mathbf{X}_l) \cdot N^{-\frac{1}{3}})$$
$$\mathbf{X}_{l+1} = \mathbf{X}_l + \text{MaxNormActivation}(\text{FFN}(\mathbf{X}'_l))$$

## 3.2 LOCALATTENTION FOR DENSEATTENTION

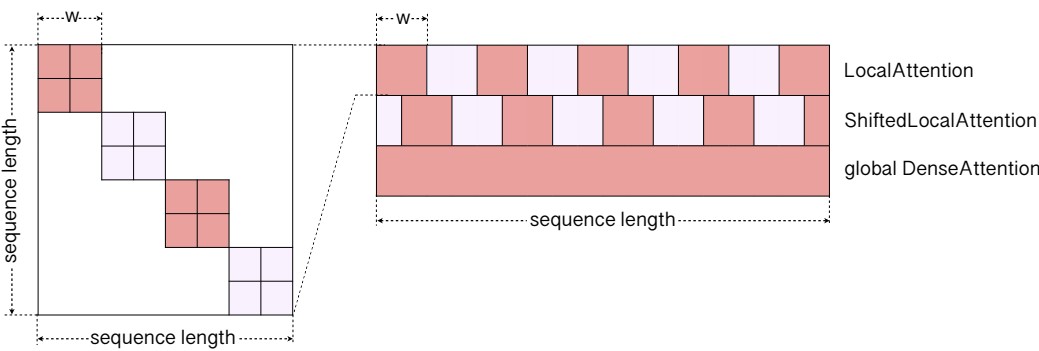

Figure 3: Local attention for DenseAttention scheme. Left: Chunked attention pattern of an individual local attention layer. Right: 3 layer structure of Local – LocalShifted – global attentions.

In the years following invention of Transformer, many variations of *local attention*, also known as *sliding window attention*, patterns and implementations have been proposed (Zaheer et al., 2020; Beltagy et al., 2020; Child et al., 2019; Roy et al., 2021; Dao et al., 2022b; Xiong et al., 2022). Recently, some of the open-weights Large Language Models (Jiang et al., 2023; Gemma Team et al., 2024) started partially or fully adopting some forms of local attention with the primary goal of alleviating quadratic cost of full attention for large contexts with the trade-off of not being able to fully process the entire sequence at once.

We also develop a form of local attention pattern for discretionary use with DenseAttention on very long contexts, however, with the goal of improving modeling quality as opposed to increasing speed. The reason of this extension is outlined by Qin et al. (2022a): in linear Transformer family of models, attention scores of a query are distributed along the sequence length more uniformly as compared to Softmax attention, so the model is not fully able to focus at details in the vicinity of a query's token.

We adopt the approach to partition the whole sequence into equal non-overlapping chunks of *window size $w$*, similar to Dao et al. (2022b); Qin et al. (2022a). We choose this design because of its simplicity and straight-forward implementation with minimal invocations of memory-intensive data layouts. However, this form of chunked attention leads to all of the tokens not being able to interact with up to a half of the tokens constituting their neighbourhood. To mitigate this issue, we extend our local attention framework beyond one layer and propose a 3-layer structure (Fig. 3). It consists of LocalAttention, ShiftedLocalAttention, and global DenseAttention layers. The second,

Table 1: The Long Range Arena performance. Accuracy is the metrics for all benchmarks. Best results are in bold. Full comparisons with 20+ other models are available in Table 13 (trends hold).

| Models | Listops | Text | Retrieval | Image | Pathfinder | PathX | Avg. |
|---|---|---|---|---|---|---|---|
| Transformers + Rotary | 47.90 | 79.08 | 82.31 | 75.04 | 76.64 | 84.72 | 74.28 |
| S4-v1 | **58.35** | 76.02 | 87.09 | **87.26** | 86.05 | 88.10 | **80.48** |
| DenseAttention | 50.50 | **81.19** | **87.51** | 72.55 | **87.40** | **88.82** | 77.99 |

ShiftedLocalAttention layer is shifted by $w/2$ relative to the first, which allows for all tokens to have symmetric neighbourhood after two consecutive layers. The full global attention of the last layer in the scheme combines fine-grained local results to capture all context of a sequence. The triples of layers then may be stacked together like ordinary Transformer layers to form a deep network.

We find local attention to be very effective in our experiments (as discussed in detail in 4.1 and B.3).

## 4 EXPERIMENTS

To prove the viability of DenseAttention architecture, we conduct several sets of experiments: (i) long range sequence modeling on Long Range Arena benchmark and PathFinder-256 benchmark with 64K context size; (ii) language modeling pre-training of BERT-like encoder and Llama-like decoder architectures on sequences of different lengths and scaling laws on small to moderate model sizes; (iii) speed benchmarking. We report results for BERT-pretraining on 1k and 16K context sizes, LM scaling laws of DenseAttention, and Pathfinder-256 in Appendix B. Details for all experiments are provided in Appendix G.

### 4.1 LONG RANGE ARENA

Long Range Arena is a challenging suite of 6 classification benchmarks dedicated to examining the abilities of efficient and long-context models on large sequence lengths spanning from 1k to 16k tokens. The tasks are diverse in nature and modalities: from synthetic and purely algorithmic, such as long version of ListOps benchmark (Nangia and Bowman, 2018), to text ranking/ matching and character-level text classification on IMDB reviews (Maas et al., 2011). At the time of publication, the best model tested by Tay et al. (2021) achieved average of 55.01%, and all of the models failed to learn above the level of change on the most difficult task, Pathfinder-X (seq. len 16K) adopted from Linsley et al. (2018); Kim* et al. (2020).

Later, novel State-Space-Models-inspired architectures (Gu et al., 2022a;b; Ma et al., 2023; 2024) demonstrated by far superior performance considered to be out of reach for any Transformer-based model due specific inductive biases of the SSMs. But recently, Amos et al. (2024) showed that by using MLM-style pre-training the Transformer with RoPE is competitive with the SSMs. Interestingly, even without pre-training, but with RoPE, they reached a SOTA score on the benchmark among all Transformer-based architectures with a large margin.

We take their scores as well as results from original S4 paper (Gu et al., 2022a) as two strong baselines and conduct extensive experiments on LRA dataset with DANet model to see if our architecture is capable of matching or surpassing them. We mostly follow specifications outlined in the original LRA paper including number of heads and model dimensions, adjusting where necessary number of parameters to the one used by Amos et al. (2024). We report the results in Table 1. DenseAttention Network establishes new SOTA score among the Transformer-based models and even outperforms the SSM in 4 out of 6 benchmarks. Thus, we prove that DenseAttention architecture is competitive with standard attention even despite the simplifications, the absence of Softmax, and the presence of non-smooth functions in the DANet architecture (MaxNorm and ReLU). Our results also indicate that Transformer-based models can potentially match the performance of SSMs without MLM-style pretraining as in Amos et al. (2024).

We use computationally efficient alternative to RoPE (Su et al., 2024) embeddings, Cosine RelPE (see Appendix F for implementation details), and Local–ShiftedLocal–Global attention scheme in all of

Table 2: Ablations on the Retrieval task of LRA. In- Table 3: Ablation on number of heads in tegration of the Local–ShiftedLocal–Global attention DANet-BERT model. Clearly, reducing num- scheme results in the most pronounced accuracy gain. ber of heads in DenseAttention leads to improved performance.

| Model | Accuracy |
|---|---|
| DANet + Sinusoidal Embedding (bf16 format) | 82.69 |
| DANet + Cosine RelPE | 83.98 |
| DANet + Cosine RelPE + local attention (w=10) | **87.51** |

| Model | MLM loss | Acc. |
|---|---|---|
| DANet-BERT, 1 head | **1.564** | **67.1** |
| DANet-BERT, 4 heads | 1.627 | 66.3 |
| DANet-BERT, 16 heads | 1.691 | 65.1 |

LRA models. These extensions are useful for improving results which is exemplified in Table 2. Local attention proves to be instrumental and, often, its window size is the most important hyperparameter to tune.

## 4.2 LANGUAGE MODELING

Table 4: Comparison of DANet with Transformer (upper plane) and various sub-quadratic (bottom plane) architectures across GLUE tasks. All scores are taken from their respective papers. CoLA is measured by Matthew's correlation, STS-B by Spearman's correlation, and other tasks by accuracy. QQP and MRPC are measured by F1 score for some of the models.

| Model | MNLI Acc. | QNLI Acc. | QQP Acc. | SST2 Acc. | RTE Acc. | CoLA Matthew | STS-B Spearman | MRPC Acc. | Avg. |
|---|---|---|---|---|---|---|---|---|---|
| BERT-Large (BookC+Wiki) (Liu et al., 2019) | 86.6 | 92.3 | 91.3 | 93.2 | 70.4 | 60.6 | 90.0 | 88.0 | 84.1 |
| BERT-Large (Portes et al., 2023) | 86.3 | 92.8 | 90.9 | 93.3 | 83.8 | 56.2 | 90.6 | 87.8 | 85.2 |
| MosaicBERT-L (Portes et al., 2023) | 86.9 | 93.0 | 92.0 | 93.7 | 84.5 | 59.7 | 90.9 | 88.2 | 86.1 |
| DANet-BERT-L (ours) | 87.1 | 91.9 | 91.6 | 95.0 | 84.8 | 63.0 | 89.1 | 89.0 | **86.4** |
| | Acc. | Acc. | F1 | Acc. | Acc. | Matthew | Spearman | F1 | |
| Linear Attention (Lee-Thorp et al., 2022b) | 35.0 | 84.0 | 84.0 | 79.0 | 60.0 | 67.0 | 24.0 | 73.0 | 59.8 |
| FNet (Lee-Thorp et al., 2022b) | 79.0 | 89.0 | 87.0 | 92.0 | 70.0 | 81.0 | 88.0 | 86.0 | 83.6 |
| Monarch Mixer Fu et al. (2023a) | 82.2 | 87.0 | 87.7 | 92.4 | 75.0 | 59.6 | 88.3 | 90.1 | 82.8 |
| BiGS (Wang et al., 2023) | 86.2 | 90.9 | 88.3 | 94.6 | 79.4 | 67.3 | 90.1 | 89.5 | 85.8 |
| DANet-BERT-L (ours) | 87.1 | 91.9 | 88.9 | 95.0 | 84.8 | 63.0 | 89.1 | 92.0 | **86.5** |

To validate DenseAttention capabilities in language modeling, we conduct experiments with both Masked and Causal LM, with emphasis on the former. We set number of parameters to approx. 335M and 360M parameters respectively for the two modalities, with model dimension $d = 1024$, the same hyperparameters as in BERT-Large (Devlin et al., 2019) and GPT-2 Medium (Radford et al., 2019). Further details are reported in Appendix G.

**Analysis.** Major differences between our models and Transformer baselines lie in the number of layers and heads. Since a Transformer layer has 4/3 times more parameters than DANet layer for matching model dimension, we increase number of layers from 24 to 32 to keep parity in number of parameters. As proposed in 3.1, we use a single head with the dimension equal to full model dimension ($d = 1024$), opposed to customary 16 small heads in 360M Transformers. We justify empirically (Table 3) that this choice is optimal.

We conducted evaluation of downstream language modeling capabilities on GLUE suite of benchmarks (Wang et al., 2018). In all comparisons (Table 4), DANet architecture is highly competitive, surpassing the performance of standard Transformers and various efficient sub-quadratic sequence-mixer architectures. Notably, it outperforms MosaicBERT which has considerably more parameters (430M) and modern improvements of Transformer architecture (SwiGLU FFN and ALiBi relative positional embeddings). The results in raw causal language modeling (Table 5) also indicate that DenseAttention architecture is indeed capable of achieving similar or better performance as standard Transformer, despite having nothing but dense MatMuls in attention. Based on both language modeling and LRA results, we conclude that *MatMuls are enough* for performant and efficient linear time and space attention mechanism.

**Speed and efficiency.** We compare inference (Table 6) and training (Table 8) speed of DANet to standard softmax attention Transformer as in Devlin et al. (2019) augmented with low-level

Table 5: Comparison of perfor-Table 6: Inference throughput (thousands tokens per second) compar-mance of DANet and Trans-ison across varying context sizes (in tokens) and hardware types. Best former models in CLM set-results per row group are shown in bold. OOM = Out of Memory at tings after training on 11B to-batch size 1. DANet is Pareto-optimal for all sizes and accelerators. kens from The Pile dataset (Gao et al., 2020) on full test set.

| Model | Ppl. | Acc. |
|-------|------|------|
| Llama | 8.88 | 56.6 |
| DANet-Llama | **8.79** | **56.8** |

| Model (Hardware) | 128 | 1024 | 4096 | 16384 | 65536 | 131072 |
|------------------|-----|------|------|-------|-------|--------|
| Transformer (H100) | 736.05 | 571.39 | 318.46 | 116.74 | 33.29 | 16.87 |
| Linear Attention (H100) | 563.37 | 568.19 | 568.07 | 566.95 | 566.62 | 565.84 |
| **DANet (H100)** | **772.03** | **699.60** | **701.93** | **700.73** | **697.89** | **690.36** |
| Transformer (A100) | 303.62 | 257.54 | 165.46 | 68.04 | 20.27 | 10.47 |
| Linear Attention (A100) | 243.72 | 241.66 | 242.81 | 241.65 | 243.39 | 242.73 |
| **DANet (A100)** | **313.25** | **277.52** | **277.71** | **277.92** | **273.71** | **272.96** |
| Transformer (CPU) | 7.99 | 2.21 | 0.62 | 0.16 | OOM | OOM |
| Linear Attention (CPU) | 7.67 | 7.75 | 7.67 | 7.73 | 7.75 | 7.82 |
| **DANet (CPU)** | **14.97** | **13.60** | **13.21** | **12.94** | **13.46** | **12.83** |

FlashAttention optimizations (Dao, 2024; Shah et al., 2024), and to Linear Transformer with $1 + \mathrm{elu}$ kernel from Katharopoulos et al. (2020). All models have approx. 340M parameters (BERT-Large size) and are used with PyTorch 2.x compiled mode. DANet and Linear Transformer use 'auto' switch to choose run-time complexity depending on the sequence length (see Appendix B.1 for an extended exploration of DANet's advantageous *computational efficiency* even in fixed $O(N^2)$ mode). Accelerators used in these experiments are 1x NVIDIA H100 80GB HBM3, 1x NVIDIA A100-SXM4-80GB, and 12 cores of Intel Xeon Platinum 8480+ CPU with 364 GB DDR5 RAM.

We find that DANet is invariably faster than both alternatives on each of the accelerators for all context sizes, from very short (128 tokens) to extremely long (131k tokens). Particularly, DANet consistently Pareto-improves on the Linear Transformer for every sequence length and on every tested device. This is especially noteworthy as Linear Transformer is among the strongest counterparts to DenseAttention architecture in terms of computational efficiency because (i) it has computational complexity $O(N)$ as opposed to $O(N \log N)$ for State Space Models like Gu et al. (2022b); Gupta et al. (2022) and FFT-based architectures like Lee-Thorp et al. (2022a); Fu et al. (2023b); (ii) it has one of the simplest algorithms and relatively few non-linear and element-wise transformations among the large class of linear-time algorithms, which makes it one of the fastest $O(N)$ models.

DANet's advantage in speed is most notable on CPU, where specialized kernels like FlashAttention are not available. In this setting, DANet achieves a minimum speed-up of $2\times$ over Transformer, starting at the smallest context size. Yet even on NVIDIA H100, DANet is faster than FlashAttention by 22.4% on small 1024-token context size and by orders of magnitude on large sequences, indicating superior computational efficiency of DenseAttention.

## 5 CONCLUSION

In this paper, we propose DenseAttention Network – a general architecture which simplifies the Transformer block and can serve as a drop-in replacement in every model architecture using it. We conduct experiments on the diverse modalities spanning from language modeling and NLP tasks to logic and image classification, and from short to extremely long sequence lengths up to 64K tokens, using the LRA and GLUE suites of benchmarks, and MLM and CLM style language model pre-training on text data. The results show that DenseAttention is capable of generalizing to many different tasks and context sizes and achieving favorable performance in comparison with standard Transformer and its augmented variants.

Both linear-time complexity and improved computational efficiency lead to greater speeds of DANet architecture, compared to other linear-time models and Transformer augmented with specialized, low-level computation algorithms such as in Dao et al. (2022b). Since the architecture relies only on PyTorch and works seamlessly with an older fp16 precision format, it can be used on a wide range of CPUs, old generations of NVIDIA GPUs, and non-NVIDIA GPUs without additional modifications, democratizing the community access to modern linear-time sequence-modeling architectures.

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

CONTENTS

## A  RELATED WORK: SUB-QUADRATIC ALGORITHMS FOR SEQUENCE PROCESSING

**Linear Attention class.**  Given entries $Q_i, K_j, V_j \in \mathbb{R}^{1 \times d}$ of matrices $\mathbf{Q}, \mathbf{K}$ and $\mathbf{V}$, standard softmax attention for input $i$ can be reformulated as

$$A_i = \frac{\sum_{j=1}^{N} \mathrm{Sim}(Q_i, K_j) V_j}{\sum_{j=1}^{N} \mathrm{Sim}(Q_i, K_j)} \in \mathbb{R}^{1 \times d},$$

where $\mathrm{Sim}(Q_i, K_j) = \exp(Q_i K_j^{\top})$. Conceptually, linear attention class of algorithms, described in Katharopoulos et al. (2020) and built upon in numerous subsequent works, approximates or replaces this similarity function with separable kernel $\mathrm{Sim}(Q_i, K_j) = \mathcal{K}(Q_i, K_j) = \phi(Q_i)\phi(K_j^{\top})$, where $\phi : \mathbb{R}^d \to \mathbb{R}_+^r$ maps query and key vectors to non-negative vectors with possibly different dimension $r$.

Hence, the attention mechanism becomes:

$$\begin{aligned}
A_i &= \frac{\sum_{j=1}^{N} \phi(Q_i)\phi(K_j^{\top})V_j}{\sum_{j=1}^{N} \phi(Q_i)\phi(K_j^{\top})} \\
&= \frac{\phi(Q_i) \sum_{j=1}^{N} \phi(K_j^{\top})V_j}{\phi(Q_i) \sum_{j=1}^{N} \phi(K_j^{\top})},
\end{aligned} \tag{5}$$

which can be computed in linear time.

The function $\phi(\cdot)$ can take various forms, such as 1 + ELU (Katharopoulos et al., 2020), ReLU (Qin et al., 2022b), squared ReLU (Hua et al., 2022), Taylor (Duman Keles et al., 2023; Arora et al., 2024; Aksenov et al., 2024; Zhang et al., 2024) or Random Feature (Choromanski et al., 2022; Peng et al., 2021) expansions, and even MLPs trained to mimic softmax attention (Zhang et al., 2024). They aim to approximate softmax without its explicit calculation when being applied jointly to queries and keys, or to retain its properties, most importantly, non-negativity of resulting dot products $\phi(Q_i)\phi(K_j^{\top})$.

The latter property, together with reweighting attention scores (denominator in the formula equation 5) are *defining* for Linear Transformer algorithms. Absence of scaling by $\frac{1}{\phi(Q_i) \sum_{j=1}^{n} \phi(K_j^{\top})}$ leads to numerical instabilities, and the scaling factor itself is not guaranteed to be bounded without non-negative $\phi(\cdot)$. However, both mappings $\phi(\cdot)$, and memory intensive non-MatMul operations for reweighting contribute to subpar speed and computational efficiency in comparison with ordinary and fast self-attention algorithms on all but large context sizes.

DenseAttention is substantially different from LinearTransformers. We forgo both transforming $\mathbf{Q}, \mathbf{K}$ by $\phi(\cdot)$ and reweighting in DenseAttention as we believe the main factor of success of Transformer is the ability of all $N \times N$ interactions between tokens. It results in improved computational efficiency and simpler design which can be expressed entirely by matrix multiplications:

$$\mathbf{A} = \mathbf{Q}\mathbf{K}^{\top}\mathbf{V}$$

**SSMs and Linear RNNs.**  Another promising line of work focuses on applying deep State Space Models (SSMs) (Gu et al., 2022a; Gupta et al., 2022; Ma et al., 2023; Sun et al., 2024; Gu and Dao, 2024) and Linear RNNs (Beck et al., 2024; Orvieto et al., 2023; Peng et al., 2023) to long-range sequence and language modeling. Fundamentally, these architectures model interactions in sequence dimension by a linear recurrence:

$$\begin{aligned}
x_t &= \overline{\mathbf{A}} x_{t-1} + \overline{\mathbf{B}} u_t \\
y_t &= \overline{\mathbf{C}} x_t + \overline{\mathbf{D}} u_t,
\end{aligned}$$

where recurrence matrix $\overline{\mathbf{A}}$ and other parameters are data-independent matrices which form and initialization are defining properties for a particular SSM/ RNN architecture. The linear recurrence is advantageous during inference as it runs in $O(N)$ time. For training, it also can be unrolled into a convolutional kernel

$$\mathbf{K} = \begin{bmatrix} \overline{\mathbf{CB}}, & \overline{\mathbf{CAB}}, & \dots, & \overline{\mathbf{CA}}^{N-1}\overline{\mathbf{B}} \end{bmatrix}$$

to compute

$$y = \mathbf{K} * u$$

via Fast Fourier Transform (FFT) in $O(N \log N)$ time. Here, we set $\mathbf{D} = 0$ for ease of exposition, but in practice it's usually set to identity to act as a skip-connection ubiquitous in modern deep NN architectures.

While being sub-quadratic, these algorithms are still slower than linear time as in DenseAttention. However, recently Gu and Dao (2024); Yang et al. (2024) introduced data-dependent gating for SSM parameters and low-level, hardware efficient CUDA implementations for parallel-scan operation which allow for fast linear-time processing both during training and inference. However, it admits no training and inference without resorting to low-level implementations. This architecture type is also inherently applicable only for auto-regressive decoder-only models and is not capable of bidirectional context processing without significant modifications, such as second pass over the input sequence.

**Other algorithms.** Narrowing the focus to specific *vision* tasks, such as object detection and instance segmentation, Zhuoran et al. (2021) propose two variations of attention, one without Softmax and the other with two softmaxes applied individually to Key and Query projections. However, they conduct experiments and report results only with second architecture. Recently, Koohpayegani and Pirsiavash (2024) instead scale Queries and Keys separately by their $l_1$ norm which allows them to successfully train a vision Transformer on ImageNet1K (Deng et al., 2009) and MS-COCO (Lin et al., 2014) datasets for different tasks with linear time complexity.

MLP-Mixer (Tolstikhin et al., 2021), also tested exclusively on vision tasks, uses one primitive, two-layer MLP, for both sequence and model dimensions. In this architecture, the attention mechanism gets replaced by an MLP of fixed dimensions. This architecture has the following limitations: 1) After training, MLP parameters stay fixed, whereas in true attention mechanisms the input representations $\mathbf{X}$ interact with data-dependent matrix $\mathbf{QK^T}$; 2) It admits only sequences with fixed lengths.

Among other novel algorithms, there is an architecture class that relies notably on FFT or its generalizations such as Monarch matrices (Dao et al., 2022a). Its members include Long Convolutions (Fu et al., 2023b), Hyena (Poli et al., 2023), Monarch Mixer (Fu et al., 2023a), and FNet (Lee-Thorp et al., 2022b) with the latter also using sub-quadratic primitives both for computations along the sequence length and the model dimension.

# B  ADDITIONAL EXPERIMENTS

## B.1  MFU & SPEED COMPARISONS

We thoroughly evaluate (Table 7) DANet-BERT model inference speed, as measured by throughput, and computational efficiency by means of MFU – Model FLOPS Utilization, which shows the ratio between actual FLOPs used by the model and theoretical hardware upper limit. We compare the model with standard BERT-Large model and with highly-optimized, low-level FlashAttention-2 implementation (Dao, 2024) which is the fastest available kernel for attention computation as of early 2025. We test on sequences with lengths of all powers of 2 between 128 and 131K on a single NVIDIA A100 with 40Gb. BERT with plain PyTorch attention could not fit into GPU memory with selected batch sizes for sequences greater than 1024 tokens, as opposed to DANet in both $O(N^2)$ and $O(N)$ regimes.

All evaluations were performed using torch.compile() directive. As expected, DenseAttention model vastly outperforms both PyTorch (up to 77.2% on seq.len 1024) and even FlashAttention algorithm with either quadratic or linear regime, depending on the sequence length. For small sequences, quadratic mode of DenseAttention is preferable, and for long contexts, linear mode is indispensable in obtaining orders-of-magnitude speedups. Our architecture is also more hardware-efficient as it outperforms Transformer in terms of MFU for all context sizes. For each size, we record MFU for the

Table 7: Inference throughput (thousands tokens per second), Model FLOPs Utilization (MFU) ratio (%) of the fastest model in a group, for DenseAttention model in $O(N)$ and $O(N^2)$ regimes in comparison to BERT with and without FlashAttention-2. For all sequence lengths, DANet is faster than both FlashAttn and PyTorch BERT, as shown by speedups w.r.t respective BERT implementations. All measurements were conducted on a single NVIDIA A100 40Gb GPU.

| Seq. Len. | BS | DenseAttention | | | BERT | | | Speedup, % |
|---|---|---|---|---|---|---|---|---|
| | | $O(N)$ | $O(N^2)$ | MFU | FlashAttn-2 | PyTorch | MFU | FA-2 (PT) |
| 128 | 512 | 179.58 | **220.29** | 48.4 | 202.75 | 185.60 | 44.3 | 8.65 (18.7) |
| 512 | 256 | 204.85 | **221.08** | 52.7 | 194.30 | 156.11 | 44.8 | 13.8 (41.6) |
| 1024 | 128 | **213.91** | 213.81 | 55.0 | 185.96 | 120.73 | 45.8 | 15.0 (77.2) |
| 2048 | 64 | **197.47** | 174.08 | 50.8 | 167.30 | *OOM* | 46.6 | 18.0 |
| 4096 | 32 | **196.98** | 136.72 | 50.7 | 138.98 | *OOM* | 47.7 | 41.7 |
| 8192 | 16 | **198.08** | 96.75 | 51.0 | 102.56 | *OOM* | 48.5 | 93.1 |
| 16384 | 8 | **220.69** | 67.17 | 56.8 | 67.50 | *OOM* | 49.3 | 226.9 |
| 32768 | 4 | **197.26** | 32.28 | 50.8 | 40.11 | *OOM* | 50.0 | 391.8 |
| 65536 | 2 | **198.57** | 24.77 | 51.1 | 22.15 | *OOM* | 50.5 | 796.4 |
| 131072 | 1 | **210.24** | 11.40 | 54.1 | 11.67 | *OOM* | 50.7 | 1702 |

Table 8: Training throughput (thousands tokens per second) comparison across varying context sizes (in tokens) for pre-training on 2 Nvidia A100 80GB

| Seq. Len. | 128 | 256 | 512 | 1024 | 2048 | 4096 | 8192 |
|---|---|---|---|---|---|---|---|
| Transformer (FlashAttention) | 184.62 | 180.42 | 171.18 | 154.90 | 130.10 | 98.19 | 66.21 |
| DANet | 193.03 | 192.91 | 192.70 | 188.67 | 189.03 | 188.70 | 188.81 |
| Speed-up, % | 4.6 | 6.9 | 12.6 | 21.8 | 45.3 | 92.2 | 185.2 |

fastest regime which leads to smaller values of the metrics for long sequences. Yet, surprisingly, we also observe that with the increase of the sequence length the performance of the DenseAttention in the $O(N^2)$ regime is roughly similar to FlashAttention despite being written in high-level language and having up to $32/24 = 1.33$ times more FLOPs per iteration in the limit. It implies that DANet is up to 1.33 times more computationally efficient than Transformer.

Additionally, we conducted comparisons of training speed across different sequence lengths (Table 8), in the setting similar to the one provided in Section 4.2, except that the tests were performed using 2 Nvidia A100 GPUs simultaneously, following a real-world task of language model pre-training. The training throughputs of DANet and FlashAttention-augmented Transformer both follow the similar law as the inference speeds from Table 6.

## B.2 PATHFINDER-256

Pathfinder-256 is an extremely challenging version of the Pathfinder task with sequence length 65k which is on par with input context size of recent generations of proprietary Large Language Models.

Table 9: Accuracy on Pathfinder-256 task

| Algorithm | Acc. on val. set, % |
|---|---|
| FlashAttention (Dao et al., 2022b) | 63.1 |
| S4 (Amos et al., 2024) | 67.8 |
| DenseAttention | 72.6 |
| DenseAttention after additional 550 epochs | 77.1 |

DenseAttention model outperforms (Table 9) existing results from the literature of standard Transformer augmented with FlashAttention (Dao et al., 2022b) and S4-v2 model (Gu et al., 2022b) as reported in Amos et al. (2024). The result holds even when the training procedure is carried out for

only 200 training epochs as in Dao et al. (2022b), and further improves when it is prolonged for 550 additional epochs.

This experiment lets us make several observations:

- DenseAttention Network architecture performs well even on very long input sequences which is promising given current trend of increasing context size in modern Large Language and Multimodal Models;

- DenseAttention shows favorable scaling properties with respect to the amount of training iterations, even with the fixed dataset size. The validation accuracy for the task kept improving throughout the whole training and would likely have continued if the experiment had not been stopped;

- Truly linear scaling in sequence length is crucial for improvements in quality for large contexts. It took approximately 3 days on 4 H100 GPUs to train our model for 750 epochs in linear mode, while the projected runtime of quadratic FlashAttention-2 (Dao, 2024) and log-linear (S4) algorithms in the same setting would be at best 3 and 0.5 months, respectively, which renders them impractical for prolonged training.

### B.3    DANet-BERT for Long Contexts

Table 10: Comparison of DANet-BERT-Large pretrained on long context sizes with and without local attention. The models with context size 1k and 16k were evaluated on the corresponding length texts from C4 and Bookcorpus (held-out split) datasets respectively. "Samples" denotes the number of sequences of corresponding length seen by a model during continual pretraining.

| Metrics | 1k | | | 16k | | |
|---|---|---|---|---|---|---|
| | Samples | MLM Loss | MLM Acc. | Samples | MLM Loss | MLM Acc. |
| DANet-BERT | 80M | 2.255 | 0.591 | 27M | 2.843 | 0.452 |
| DANet-BERT + local attention | 80M | **1.705** | **0.647** | 7.8M | **1.689** | **0.637** |

We conducted additional experiments by taking DANet-BERT model after it had finished pre-training on sequence length 512 and continued pre-training on sequence lengths 1024 and then 16384 tokens both with and without local attention scheme. The results (Table 10) show that introduction of local-global attention pattern helps to quickly recover the modeling performance even on extremely long sequences. It brings the performance to the same level we observed when pre-training on small sequences and significantly outperforms the models which were pre-trained without the local attention.

Moreover, we observe that quality evaluation metrics stay the same for a fixed length validation context if the regime gets switched from $O(N)$ to $O(N^2)$ or vice versa regardless of the mode and sequence length with which a DenseAttention model has been trained. This invariance property holds even for the model trained on 16k context and applied to sequence length 128. Thus, we can train the models with DenseAttention on very large contexts in $O(N)$ time and then use it for both short and long sequences with optimal speed and equal quality.

### B.4    Scaling Effects Study

Table 11:  Scaling study on DenseAttention-BERT architecture

| Model | Parameters | Configuration | MLM loss | MLM accuracy |
|---|---|---|---|---|
| DANet-BERT-small | 31M | L=6, D=512 | 2.74 | 49.5 |
| DANet-BERT-base | 110M | L=16, D=768 | 2.02 | 60.0 |
| DANet-BERT-large | 336M | L=32, D=1024 | 1.70 | 64.9 |

The Table 11 details three single-head DenseAttention Network models of different sizes pre-trained on Wiki+BookCorpus dataset with MLM objective for 100B tokens. MLM loss and accuracy are reported for out-of-sample data from C4 dataset (Raffel et al., 2019). L and D parameters denote number of layers and hidden dimension of FFN input, respectively. DenseAttention architecture exhibits favorable scaling properties similar to standard Transformer.

## B.5 ANALYSIS OF MAXNORM AND LAYERNORM

Both $l_2$ and $l_\infty$ norms, providing mathematical foundation for LayerNorm and MaxNorm respectively, are bounded by each other up to a constant non-negative factor. However, in practical settings, this constant factor has a great impact on the range of values produced by applying either of the two norms. Consider a d-dimensional 0-mean vector $\mathbf{x}$, s.t. $\|\mathbf{x}\|_\infty = a$. Then:

$$\|\mathbf{x}\|_\infty < \|\mathbf{x}\|_2 \le \sqrt{d}\|\mathbf{x}\|_\infty \quad \text{(equivalence of norms)},$$

$$\frac{1}{\sqrt{d}\|\mathbf{x}\|_\infty} \le \frac{1}{\|\mathbf{x}\|_2} < \frac{1}{\|\mathbf{x}\|_\infty} \quad \text{(reversed)},$$

$$\frac{\mathbf{x}}{\|\mathbf{x}\|_\infty} \le \frac{\sqrt{d}\,\mathbf{x}}{\|\mathbf{x}\|_2} < \frac{\sqrt{d}\,\mathbf{x}}{\|\mathbf{x}\|_\infty} \quad \text{(multiplied by } \sqrt{d}\,\mathbf{x}),$$

which is equivalent to

$$\text{MaxNorm}(\mathbf{x}) \le \text{LayerNorm}(\mathbf{x}) < \sqrt{d}\,\text{MaxNorm}(\mathbf{x}),$$

because $\text{std}(\mathbf{x}) = \frac{\|\mathbf{x}\|_2}{\sqrt{d}}$.

This means that any output element of LayerNorm can be greater than the corresponding element of MaxNorm up to $\sqrt{d}$ times. Consequently, DenseAttention outputs become up to $d^{3/2}$ times higher, which leads to numerical overflows or high absolute values with reduced numerical accuracy for half-precision formats. And if LayerNorm is scaled down by $\frac{1}{\sqrt{d}}$, its lower bound becomes $\sqrt{d}$ times less than MaxNorm, leading to attention outputs down to $d^{3/2}$ times lower than corresponding MaxNorm outputs, and causing numerical underflows. The factor $d^{3/2}$ is derived by setting $a = \sqrt{d}$ in equation 4.

To provide concrete evidence of the under/overflows, we provide a simple numerical example. For commonly used head size $d_h = 128$, maximum absolute value of the DenseAttention output in case of LayerNorm is proportional to $d^{3/2} * d = 128^{3/2} * 128 = 185,363.8$. This is significantly larger than fp16 max value of 65,504. And for the head size $d_h = 1024$, employed in our work, it becomes 33,554,432, which is 512 times greater than fp16 max. Moreover, in practice we need the max absolute value of attention outputs to be well below fp16 max, because they serve as an immediate input to FFN which further increases their magnitude. No initialization scheme could compensate for that without reducing the network's weights to 0 numerically, because fp16 minimum absolute value is $6.10 \times 10^{-5}$.

Table 12: Comparison of LayerNorm and MaxNormActivation under different half-precision formats.

| LN Type | Precision | Accuracy (%) |
|---|---|---|
| LayerNorm | fp16 | FAIL |
| MaxNorm | fp16 | **61.3** |
| LayerNorm | bf16 | 61.6 |
| MaxNorm | bf16 | **61.9** |

To confirm empirically the analysis above, we conducted an ablation study by pretraining DANet-BERT on 10B tokens of C4 dataset and evaluating on held-out data with either standard LayerNorm

(Ba et al., 2016) or MaxNormActivation in fp16 and bf16 half-precision formats. We report the results in Table 12. We confirm that DenseAttention is not compatible with LayerNorm in older fp16 format, since the training becomes unstable and diverges early after launch, as previously noted in section 3.1 and explained in this section. We also observe that MaxNorm is optimal for bf16 training.

## C  HARDWARE EFFICIENCY

All calculations performed by a hardware accelerator such as a NVIDIA GPU are either compute-bound or memory-bound (Williams et al., 2009). It depends on whether the operation in question spends the majority of time directly on computation or on data movements between High-Bandwidth Memory (HBM) and processing units. Customary unit of measurement for computational performance is TeraFLOPs (TFLOPs) per second and for memory it's bandwidth (throughput) in TB/s. Arithmetic intensity unifies both and is calculated as $\frac{\text{number of FLOPs}}{\text{number of bytes accessed}}$. It can be attributed both to hardware accelerator (usually referred to as *ops:byte ratio* in this case) and to a computational kernel, e.g. layer of neural network, and it's necessary but not sufficient for the kernel to maintain the arithmetic intensity higher than the accelerator in order to be computationally intensive (NVIDIA Docs, 2023a). Otherwise, processing units stay idle part of the time waiting for the data to be brought from or written to HBM.

In latest generations of GPUs, FLOPs count rapidly grows but memory bandwidth progression falls behind, which results in latest generations of GPUs having much higher arithmetic intensity. Thus, it's increasingly hard for existing Deep Learning (DL) primitives to achieve hardware efficiency. Most operations besides matrix-matrix multiplications are inherently memory limited even on older GPUs. For example, the arithm. intensity of ReLU is 0.25 FLOPS/B, and for LayerNorm it's < 10 FLOPS/B on NVIDIA V100 as stated in NVIDIA Docs (2023b). Moreover, GPUs feature fast Tensor Cores (312 TFLOPs for half-precision formats in NVIDIA A100) specialized for matrix multiplications, and general purpose cores with significantly lower throughput (19.5 TFLOPS in NVIDIA A100) which in turn process non-MatMul operations even slower as reported in He (2022).

Therefore, from the view of computational efficiency, all activations, elementwise operations and reductions are detrimental to high ratios of hardware utilization, and it's beneficial to eliminate most, if not all of them. An ideal algorithm should contain merely matrix multiplications with no activations, normalizations and residual connections. However, while theoretically possible to construct a non-linear decision boundary with just a composition of matrix multiplications (e.g., by taking an input matrix $\mathbf{X}$ to a natural power), it remains a challenging task in practice due to numerical instabilities occurring both in forward and backward pass and lagging performance of such architectures (Balduzzi et al., 2017; Santurkar et al., 2018; Pascanu et al., 2013).

## D  DISSECTING INEFFICIENCIES IN TRANSFORMER

Non-linearities, namely Softmax, LayerNorms, activation in FFN, dropouts, and skip-connections, which are present in Transformer architecture, indeed contribute majorly to its computational inefficiency, as documented in Ivanov et al. (2021); Pati et al. (2022); Portes et al. (2023). But other affine or linear transformations might also require further exploration. Consider two matrices $\mathbf{A} \in \mathbb{R}^{M \times N}$ and $\mathbf{B} \in \mathbb{R}^{N \times K}$ stored in half-precision floating point format which is common for DL applications. Each element in the matrices has a size of 2 bytes, and each fused multiply-add (FMA) operation takes 2 FLOPs to compute (NVIDIA Docs, 2023a). Then the arithmetic intensity of matrix multiplication in such setting is:

$$\frac{M \cdot N \cdot K}{M \cdot N + N \cdot K + M \cdot K} \text{ FLOPs/B,} \tag{6}$$

as factors of 2 in the numerator and denominator both cancel out.

If there are no biases, then the two linear transformations in Transformer's FFN with model dimension $d$ and standard inner dimension $4d$ have arithm. int. of $\frac{4Nd}{5N+4d}$ which equals $\frac{4d}{5}$ as $N \to \infty$. $N$ dimension can accumulate both batch size $b$ and sequence length $s$ dimensions, and for BERT-large size model with $d = 1024, s = 512$, and $b = 128$ arithm. int. is approx. 809 FLOPs/B. For largest LLaMA 2 70B model with $d = 8192, s = 4096$, and $b = 1$ theoretical arithm. int. without using tensor parallelism (Narayanan et al., 2021) would be 2520 FLOPs/B. It's far greater than even NVIDIA

H100 ops:byte ratio in both cases. Therefore, linear layers in the FFN are the most computationally efficient component of the Transformer and should be preserved in any hardware-aware architecture.

Similar argument may be applied to $K, Q, V$ projection layers in the self-attention, whose matrices can be concatenated together to yield $\frac{3d}{4}$ asymptotic arithm. intensity, and to the output projection by $W_O$ matrix in equation 2 ($\frac{d}{2}$ asymptotic arithm. int.). However, it follows from equation 6 that both products $\mathbf{S} = \mathbf{Q}\mathbf{K}^{\top} \in \mathbb{R}^{N \times N}$, and $\mathbf{O} = \mathbf{P}\mathbf{V} \in \mathbb{R}^{N \times d}$, where $\mathbf{P} = \text{Softmax}(\mathbf{S} \, / \, \sqrt{d_h} + \mathbf{M})$ have arithmetic intensity $\frac{N \cdot d_h}{N + 2d_h}$ with limit $d_h$ when $N \to \infty$. Also, batch and sequence dimensions cannot be fused for these operations because they are performed on *per sequence* level as opposed to *per embedding* level in FFN and KQV projections.

Large number of attention heads also contributes to inefficiency. Projection dimension of a head $i$ $\mathbf{Q}_i, \mathbf{K}_i$, and $\mathbf{V}_i$ is $\frac{d}{h}$ and typically equals 64 for smaller NLP language models like BERT, 256 for Google's PaLM (Chowdhery et al., 2022), and 128 for most others in the billions-parameters range, including LLAMA model family (Touvron et al., 2023a;b), Mistral (Jiang et al., 2023) and Mixtral 8x7B (Jiang et al., 2024), and GPT-3 (Brown et al., 2020).

Since the most common choice for $d_h$ is 128, the upper bound of arithm. int. of matrix multiplications inside attention mechanism is lower than even *ops:byte ratio* of an older V100 generation GPU. In the case of real-life configurations of BERT and LLaMA 2 from above the values are 32 and 120.5 FLOPs/B correspondingly. Thus, these operations are memory-bound and inefficient.

As extensive research efforts have shown (Bhojanapalli et al., 2020; Voita et al., 2019; Kovaleva et al., 2019; Michel et al., 2019), significant portion of heads in multi-head attention are redundant, output low-rank representations and can be pruned without decrease in quality in downstream tasks, at least for BERT-sized models. Specifically, Bhojanapalli et al. (2020) find that increasing number of heads past a certain threshold degrades performance in BERT.

So, from the computational and qualitative perspectives, it is beneficial to change the number of heads in the attention to fewer or even a single head with larger dimension $d_h$. Furthermore, it keeps the total number of flops constant because it equals $h \cdot N^2 \frac{d}{h} = N^2 d$ for all heads in total. For example, increasing $d_h$ from conventional value 128 up to 1024, in case of BERT from the example above would lead to a single-head attention with arithm. int. 204.8 FLOPs/B which makes it computationally efficient even on NVIDIA A100. For LLMs with larger model dimension $d_h = 1024$ would still leave room for multiple heads. And asymptotic arithm. int. in $O(N)$-regime is $\frac{d}{2}$ just like in an ordinary $d \times d$ dense layer.

# E    THE LRA BENCHMARK

## E.1    DISCUSSION OF THE LRA TASKS

The Long Range Arena is a suite of 6 challenging and diverse tasks designed to test modeling capabilities across different domains. Below is a brief description of each task.

**ListOps** (Nangia and Bowman, 2018). This is a purely logical synthetic task which is dedicated to modeling evaluation results of long hierarchically structured sequences. Each sequence has length up to 2000 symbols and consists of whole numbers from 0 to 9, mathematical operators, such as MAX, MIN, MEDIAN and SUM_MOD, and parentheses.

**Text Classification (IMDB)** (Maas et al., 2011). This task tests Natural Language Understanding (NLU) abilities of models by letting them classify the sentiment of movie reviews in the IMDB dataset. To make the task more challenging, the texts of the reviews are split into tokens not on a word level, but on a character (or byte) level. This leads to much longer sequences of 4K max length.

**Document Retrieval (AAN)** (Radev et al., 2013). This task tests the abilities of producing encoded representations of the textual information and further matching/ retrieving them. Namely, given a pair of the documents from ACL Anthology Network (AAN; Radev et al., 2013) dataset, a model should independently process them and, based on their final embeddings, classify if the two documents have a citation link. As in the IMDB tasks, individual input texts are tokenized on a character (byte) level with maximum sequence length of 4K.

Table 13: Long Range Arena performance. Accuracy is the metrics for all benchmarks. Best results are in bold and second best are underscored. To ensure consistent comparisons, the averages for the models which report the result on Path-X task are computed without it.

| Model | Listops | Text | Retrieval | Image | Pathfinder | PathX | Avg. |
|---|---|---|---|---|---|---|---|
| Transformer (Tay et al., 2021; Dao et al., 2022b) | 36.37 | 64.27 | 57.46 | 42.44 | 71.40 | 61.40 | 54.39 |
| Local Attention (Tay et al., 2021) | 15.82 | 52.98 | 53.39 | 41.46 | 66.63 | - | 46.06 |
| Sparse Trans. (Tay et al., 2021) | 17.07 | 63.58 | 59.59 | 44.24 | 71.71 | - | 51.24 |
| Longformer (Tay et al., 2021) | 35.63 | 62.85 | 56.89 | 42.22 | 69.71 | - | 53.46 |
| Linformer (Tay et al., 2021) | 35.70 | 53.94 | 52.27 | 38.56 | 76.34 | - | 51.36 |
| Reformer (Tay et al., 2021) | 37.27 | 56.10 | 53.40 | 38.07 | 68.50 | - | 50.67 |
| Sinkhorn Trans. (Tay et al., 2021) | 33.67 | 61.20 | 53.83 | 41.23 | 67.45 | - | 51.29 |
| Synthesizer (Tay et al., 2021) | 36.99 | 61.68 | 54.67 | 41.61 | 69.45 | - | 52.88 |
| BigBird (Tay et al., 2021) | 36.05 | 64.02 | 59.29 | 40.83 | 74.87 | - | 55.01 |
| Linear Transformer (Tay et al., 2021) | 16.13 | 65.90 | 53.09 | 42.34 | 75.30 | - | 50.55 |
| Performer (Tay et al., 2021) | 18.01 | 65.40 | 53.82 | 42.77 | 77.05 | - | 51.41 |
| RFA (Peng et al., 2021) | 36.80 | 66.00 | 56.10 | - | - | - | - |
| Luna-256 (Ma et al., 2021) | 37.98 | 65.78 | 79.56 | 47.86 | 78.55 | - | 61.95 |
| Nyströmformer (Xiong et al., 2021) | 37.15 | 65.52 | 79.56 | 41.58 | 70.94 | - | 58.95 |
| Kernelized Attention (Chen et al., 2021) | 38.78 | 60.22 | 81.77 | 41.29 | 70.73 | - | 58.56 |
| Informer (Chen et al., 2021) | 32.53 | 62.64 | 77.57 | 38.10 | 57.83 | - | 53.73 |
| Skyformer (Chen et al., 2021) | 38.69 | 64.70 | 82.06 | 40.77 | 70.73 | - | 59.39 |
| cosFormer (Qin et al., 2022b) | 37.90 | 63.41 | 61.36 | 43.17 | 70.33 | - | 55.23 |
| FNet (Lee-Thorp et al., 2022b) | 35.33 | 65.11 | 59.61 | 38.67 | 77.80 | - | 55.30 |
| FLASH-quad (Qin et al., 2022a) | 42.20 | 64.10 | 83.00 | 48.30 | 63.28 | - | 60.18 |
| FLASH (Qin et al., 2022a) | 38.70 | 64.10 | 86.10 | 47.40 | 70.25 | - | 61.31 |
| TransNormer T1 (Qin et al., 2022a) | 41.03 | 66.90 | 83.11 | 51.60 | 75.92 | - | 63.71 |
| TransNormer T2 (Qin et al., 2022a) | 41.60 | 72.20 | 83.82 | 49.60 | 76.80 | - | 64.80 |
| KDEformer (Zandieh et al., 2023) | 36.64 | 62.00 | 73.52 | 45.45 | 68.13 | - | 57.15 |
| Hedgehog (Zhang et al., 2024) | 37.15 | 64.60 | 82.24 | 40.15 | 74.16 | - | 59.66 |
| Transformers + Rotary (Amos et al., 2024) | 47.90 | 79.08 | 82.31 | **75.04** | 76.64 | 84.72 | 72.89 |
| DenseAttention (ours) | **50.50** | **81.19** | **87.51** | 72.55 | **87.40** | **88.82** | **75.83** |

**Image Classification (CIFAR-10)** (Krizhevsky and Hinton, 2009). This is an image classification task with 10 classes on a classical CIFAR-10 benchmark with one specific condition: images should be ingested into models as 1-d sequences, thus setting the input length to 1024 tokens (pixels) and making the task more challenging.

**Pathfinder** (Kim* et al., 2020) . This is a binary classification task of 32x32 pixels grayscale images with corresponding sequence length 1024 tokens, which, formally, makes it similar to CIFAR-10 task. However, it's different on a conceptual level, as the task measures a model's ability to discern spatial dependencies. Given a multitude of intertwined, dashed line paths, a model should correctly determine if two rounded dots are connected by a dashed line.

**Pathfinder-X (Pathfinder-128)**. It's a version of Pathfinder task with 16K (128x128) pixels images which makes it significantly more challenging. At the time of publication of the original LRA paper Tay et al. (2021), none of the tested models managed to achieve a score above chance on this benchmark.

Therefore, the Long Range Arena arguably represents a wide range of tasks, spanning from logic and reasoning to language modeling and image classification. To perform well on all of the 6 benchmarks, a model's architecture should be powerful and versatile enough to generalize to different modalities.

### E.2 EXTENDED COMPARISONS WITH TRANSFORMER-BASED MODELS

Full comparisons with an exhaustive list of Transformer-based models which, to the best of our knowledge, have been tested on the LRA up to late 2024, including the most recent ones are presented in Table 13. The results show that DenseAttention outperforms all of the tested models.

## F    COSINE RELPE

Many modern Language Models use (Minaee et al., 2024) Rotary Positional Embeddings (RoPE) (Su et al., 2024) which evidently perform better than learned or sinusoidal positional embeddings and don't increase parameters count. The former two types of embeddings are applied once before the first layer and rely on skip-connections for propagating positional information to other layers in the stack. While it may be suitable for shallow networks, in deeper ones the signal gets decayed as more layers add their outputs to the residual branch. On the contrary, RoPE inject positional information into each of the Transformer layers by directly applying a transformation to the matrices $\mathbf{Q}$ and $\mathbf{K}$ which can be summarized as follows:

$$\mathbf{f}(\mathbf{x}_i, m) = \begin{bmatrix} \cos m\theta_i & -\sin m\theta_i \\ \sin m\theta_i & \cos m\theta_i \end{bmatrix} \begin{bmatrix} x_{i1} \\ x_{i2} \end{bmatrix},$$

where $\mathbf{x}_i = [x_{i1} \ x_{i2}]^T$ is a chunk $i$, $i \in \{0, \ldots, \frac{d}{2}\}$, of a vector $\mathbf{x}$ with $d$ dimensions which can be either a query $\mathbf{q}_m$ or key $\mathbf{k}_m$ with position $m$ out of $N$ in the sequence. Essentially, the transformation rotates the 2 two-dimensional vectors $\mathbf{q}'$ and $\mathbf{k}'$ with the intention to maximize their dot product when they share the same position in sequence, and decay it to zero when the positions largely differ. However, direct calculation shows that it's not always true, as the result for some fixed $i$:

$$\begin{aligned} \mathbf{f}^{\top}(\mathbf{q}', m)\mathbf{f}(\mathbf{k}', n) &= (q_1 k_1 + q_2 k_2)\cos(m-n)\theta \\ &+ (q_2 k_1 - q_1 k_2)\sin(m-n)\theta \end{aligned} \qquad (7)$$

is only guaranteed to follow the pattern in case $\mathbf{q}'$ and $\mathbf{k}'$ are collinear. The total dot product of $\mathbf{q}$ and $\mathbf{k}$ is even less benign, for in each position $i$ of the model dimension, corresponding two-dimensional vector chunk has a possibly distinctive prior angle from the origin, and $\theta_i$ is also unique by construction:

$$\theta_i = 10000^{-2i/d}, \qquad (8)$$

But Su et al. (2024) show that this parameterization leads to long-term decay in norm of attention scores with the increase of relative distance $m - n$.

Besides, RoPE are computationally inefficient as their calculation induces memory-expensive changes of tensor layout and several element-wise operations with low arithmetic intensity, separately for $\mathbf{Q}$ and $\mathbf{K}$. We notice that there exist two other transformations with more favorable efficiency properties which can be applied to scalars at individual positions $i \in \{0, \ldots, d\}$ of vectors $\mathbf{q}$ and $\mathbf{k}$ rather than paired numbers: $g_1(x_i, m) = x_i \cos m\theta_i$ and $g_2(x_i, m) = x_i(\cos m\theta_i - \sin m\theta_i)$. These produce similar expansions to equation 7:

$$\begin{aligned} g_1(q_i, m)g_1(k_i, n) &= q_i k_i \cos m\theta_i \cos n\theta_i \\ &= q_i k_i [\cos(m-n)\theta_i - \sin m\theta_i \sin n\theta_i] \\ g_2(q_i, m)g_1(k_i, n) \\ &= q_i k_i [\cos(m-n)\theta_i - \sin(m+n)\theta_i] \end{aligned}$$

We tested all three functions $\mathbf{f}$, $g_1$ and $g_2$ on LRA tasks with DenseAttention and found out that all of them impact the performance very similarly. However, when we set a constant $\theta$ for all positions in an embedding dimension, the quality dropped, adding evidence to the leading role of parameterization equation 8 in the RoPE potential.

We choose the simpler function $g_1$ as the new computationally efficient alternative to RoPE and name it *Cosine RelPE*. We use it extensively in conjunction with DenseAttention, however it can be readily applied to standard Transformer in place of RoPE.

We find that application of Cosine RelPE to $\mathbf{X}$ before DenseAttention layer, while affecting even matrix $\mathbf{X} = \mathbf{V}$ inside it, doesn't degrade the performance. Thus, we proceed with this architectural choice, which allows for one instead of two element-wise multiplications and can be further optimized by fusing with scaling factor $N^{-1/3}$.

### F.1    ABLATION STUDY ON RELPE

We performed an ablation study on the speed of Cosine RelPE and RoPE, and present the results in the Table 14. Cosine RelPE are significantly faster in both scenarios. "q, k" in the second row denotes that Cosine RelPE were applied separately to Q and K matrices like in regular RoPE.

Table 14: Ablation on RelPE. Comparison of training and inference speeds (in sequences per seconds) on the LRA's Pathfinder task.

| Model variant | Training Speed, (speed-up) | Inference Speed (speed-up) |
|---|---|---|
| Rotary Embeddings | 7025 (1.00x) | 16908 (1.00x) |
| Cosine Embeddings q,k | 10276 (1.46x) | 28467 (1.68x) |
| Cosine Embeddings | 10438 (1.49x) | 29630 (1.75x) |

## G ADDITIONAL TRAINING DETAILS

In this section, we discuss training procedures and hyperparameters for the experiments and ablations conducted in this work. Additionally, comprehensive configurations for all experiments, and recipes to reproduce them are available at [redacted for anonimity]

We code DenseAttention models in plain PyTorch (Paszke et al., 2019). We train all models using DeepSpeed (Rasley et al., 2020), with LM and Pathfinder-128 and 256 experiments in multi-node mode. We train in fp16 (LRA experiments) and bf16 (Language Modeling) precision, using the framework's native implementation which is similar to NVIDIA's AMP (Micikevicius et al., 2018). For all experiments, we use ADAM optimizer (Kingma and Ba, 2015) with decoupled weight decay modification (Loshchilov and Hutter, 2019) using parameters $\beta_1 = \beta_2 = 0.9$, if not stated otherwise. Most pre-training and finetuning workloads were conducted on machines with 2 or 4 NVIDIA H100 or A100 GPUs.

We report a median value of nine runs for GLUE and five runs for most of the LRA tasks, except for Pathfinder-128, where it's median of three runs due to computational burden. For similar reasons, all experiments and ablations in language modeling, along with Pathfinder 256 experiment, were performed exactly once. Depending on the implementation, dropout (Srivastava et al., 2014) is often used in various parts of the block, specifically after FFN and attention sub-blocks as in original Transformer, and in attention matrix before softmax as in BERT. But we don't use dropouts during the pre-training as we believe it won't slow down the convergence with a large corpora dataset typical for LLM pre-training. Besides, as noted by Clark et al. (2019), dropout in attention probabilities might be the reason of redundancies among attention heads. However, we use dropout extensively in LRA tasks, and in this setting, it's actually helpful to prevent overfitting early in the training run.

**MLM & CLM.** For Masked Language Modeling (MLM) and Causal Language Modeling (CLM), we closely follow Transformer architectures and training recipes from Devlin et al. (2019); Portes et al. (2023) and Arora et al. (2024), respectively, including tokenizers and datasets (Wikipedia and BookCorpus (Zhu et al., 2015), C4 (Raffel et al., 2019) for MLM, and The Pile (Gao et al., 2020) for CLM). Since Arora et al. (2024) use improved architecture, introduced in Llama paper (Touvron et al., 2023a), which incorporates SwiGLU activation (Shazeer, 2020) and RoPE, we also include them in DANet-Llama. To get comparable DenseAttention models, we drop-in replace Transformer layers in baselines with DANet ones. We also reproduce the baselines and compare our DANet-BERT (encoder, MLM model) and DANet-LLama (decoder, CLM model) with them. In language modeling experiments, we pre-train on sequences with lengths 128, 512, 1024, and 16K for MLM objective, and 2048 for CLM.

**Language modeling ablations**. Initial experiments with DANet-BERT architecture, including ablations on the number of heads and use of local attention, and scaling effect study were performed using architecture with learned positional embeddings and without local attention or SWiGLU FFN, as in the original BERT paper (Devlin et al., 2019). For these experiments, we also utilized Wiki+Books. All models have approx 335M parameters if not stated otherwise. For the ablation on the effect of local attention for long range performance (B.3), we used early variations of the DANet model, pre-trained with weight scaling (detailed below) and other minor architectural differences on 850M sequences of size 128 and 150M sequences of size 512 before resuming pre-training on longer sequences. For the ablation on the number of heads (Table 3), the models were pre-trained on approx. 40B tokens with sequence length 128 and evaluated on out-of-sample data from the same data mix.

**Weight scaling.** During early experiments on pre-training BERT-size models in fp16 format on long contexts without using local attention, we observed that, in order to further ensure numerical

stability, it is beneficial to scale weight matrices of FFN layers so that they have a constant $l_\infty$ norm after each optimizer step during pre-training. After pre-training, each weight can be merged with its final scaling factor so there is no additional overhead at the inference time. The choice of the norm type is motivated largely by the bounds it provides for the layer outputs as in the case with the DenseAttention layer. The scaling factor of a layer is a standalone non-trainable scalar decoupled from its corresponding weight tensor at the train time. This means that the weight itself doesn't get re-scaled constantly which would otherwise induce tug-of-war dynamics with the direction of gradient. This way, the weight also has natural proportions compared to ADAM optimizer's ((Kingma and Ba, 2015)) weight update as it would have in the absence of scaling. By employing this technique, we eliminate the need for weight decay and warmup. We also used constant learning rate $2 \times 10^{-4}$ in all such training runs.

We observed that scaling the Queries weight in the DenseAttention hinders loss convergence speed to a certain degree so we proceeded with scaling just FFN layers.

However, we found that using bf16 format without weights rescaling leads to faster convergence and proceeded with this setup for all subsequent language modeling experiments.

**LRA & Pathfinder-256.** We report main hyperparameters for these benchmarks in Table 15. Models for all tasks are trained with a linear warm-up for 5-10% of training steps, followed by a constant learning rate. The number of training epochs is 200 for the three *Pathfinder* tasks and 400 for *Image* with all of them continuously improving in accuracy till the end of training, while other tasks saturate after 40-60 epochs.

Table 15: Hyperparameters for LRA tasks. These include inner dimension of the model, number of layers, number of heads, size of expansion layer in the FFN relative to model dimension, window size, dropout in FFN, learning rate, batch size, and weight decay.

| Task | Hid. Size | Layers | Heads | FFN size | Window | DP | LR | BS | WD |
|---|---|---|---|---|---|---|---|---|---|
| Listops | 512 | 9 | 8 | 2 | 20 | 0.1 | 1e-3 | 128 | 0.1 |
| Text | 512 | 9 | 8 | 2 | 10 | 0.1 | 2e-3 | 100 | 0.1 |
| Retrieval | 128 | 6 | 4 | 4 | 10 | 0.05 | 1e-3 | 100 | 0 |
| Image | 256 | 9 | 4 | 2 | 64 | 0.05 | 1e-2 | 100 | 0.1 |
| Pathfinder | 128 | 6 | 8 | 1 | 256 | 0.05 | 1e-3 | 100 | 0.1 |
| PathX | 128 | 6 | 8 | 1 | 1024 | 0.05 | 1e-2 | 4000 | 0.1 |
| Pathfinder-256 | 128 | 6 | 8 | 1 | 1024 | 0.10 | 1e-2 | 4000 | 0.1 |

**GLUE.** In all comparisons from Table 4, the size of the models approximately corresponds to BERT-Large (330M-350M parameters), except for MosaicBERT-L (Portes et al., 2023) which has 430M parameters. All models, including DANet-BERT, were pre-trained on C4 dataset (Raffel et al., 2019), except for BERT-Large from Liu et al. (2019) which was trained on BookCorpus and Wikipedia. Similarly to MosaicBERT, we used SwiGLU FFN and relative positional embeddings (but RoPE instead of ALiBI as in Portes et al. (2023)), pre-training on batches of 4096 fixed-size 128 token sequences, with documents crossing sequence boundaries and no Next Sentence Prediction (NSP) task. We also utilize the Local–ShiftedLocal–Global scheme with window size $w = 32$ for this experiment, as we find that it improves modeling quality even for such small contexts. We pre-trained the base model on 485B tokens for approximately 15 days on 2 H100 GPUs. Learning rate schedule included 4 stages: 1) linear warmup from 5e-5 to 5e-4 for 11200 steps, 2) linear decay to 1e-4 for 470400 steps, 3) constant learning rate 1e-4 for 308700 steps (we increased Adam $\beta_2$ to 0.99 and gradually decreased weight decay from 0.1 to 0.05 and then to 0.02 during this stage), 4) linear decay to 0 for 120800 steps.

The results reported in Table 4 are for the validation set of GLUE. Following previous work (Fu et al., 2023a; Portes et al., 2023), after the end of stage 4, we fine-tune the original model initially on MNLI dataset for 3 epochs and then use the obtained checkpoint for further fine-tuning QNLI, QQP, RTE, STS-B, and MRPC tasks. Likewise, after 90100 steps of stage 4, we fine-tune the original model on SST2 for 4 epochs and then use the result for COLA task. We fine-tuned the model with the identical batch size 32 and no weight decay for all of GLUE tasks up to a maximum of 20 epochs, choosing

best results. The fine-tuning learning rates are constant with no warm-up and decay but different for each task (see Table 16). We set dropout to 0.05 for all tasks except CoLA and STS-B.

Table 16: DANet-BERT fine-tuning hyperparameters for GLUE tasks

| Task | CoLA | MNLI | MRPC | QNLI | QQP | RTE | SST-2 | STS-B |
|---|---|---|---|---|---|---|---|---|
| **Learning Rate** | 4e-5 | 1e-5 | 5e-5 | 1e-5 | 1e-5 | 2e-5 | 1e-5 | 4e-5 |
| **AdamW** $\beta_2$ | 0.99 | 0.9 | 0.98 | 0.9 | 0.98 | 0.98 | 0.9 | 0.98 |

## H  PROOFS

**Proof of Proposition 1:**

$$Y_{ij} = \sum_{n=1}^{N} \sum_{m=1}^{d} \sum_{k=1}^{d} X_{ik} W_{km} X_{mn}^{\top} X_{nj}$$

Denote $S(i; k; m; n; j) = X_{ik} W_{km} X_{mn}^{\top} X_{nj}$. Since $\mathbb{E}[W_{km}] = 0$ and $W_{km}$ is independent from $X$, $\mathbb{E}[S(i; k; m; n; j)] = 0$ and $\mathbb{E}[Y_{ij}] = \sum_{k,m,n} \mathbb{E}[S(i; k; m; n; j)] = 0$. Hence, $\text{Var}[S(i; k; m; n; j)] = \mathbb{E}[X_{ik}^2 W_{km}^2 (X_{mn}^{\top})^2 X_{nj}^2] - 0$.

As some of the indices $i, k, m, n, j$ can be the same number, there are three possible options for $\text{Var}[S(i; k; m; n; j)]$:

1. $\mathbb{E}[x_1^2 x_2^2 x_3^2]\mathbb{E}[w^2] = \sigma_X^6 \sigma_W^2$ by independence of all $x$ and $w$.
2. $\mathbb{E}[x_1^4 x_2^2]\mathbb{E}[w^2] = \mathbb{E}[x_1^4]\mathbb{E}[x_2^2]\sigma_W^2 \geq \sigma_X^6 \sigma_W^2$, because by Jensen's inequality $\mathbb{E}[g(x^2)] \geq g(\mathbb{E}[x^2])$ and we let $g(f) = f^2$.
3. $\mathbb{E}[x^6]\mathbb{E}[w^2] \geq \sigma_X^6 \sigma_W^2$ by similar reasoning ($g(f) = f^3$ is convex on $(0, \infty)$).

Finally, $Cov(S_p, S_q) = 0$ if the set of indices $p$ is not identically equal to set $q$ because even one distinct index between $p$ and $q$ leads to independent factors inside the covariance operator. Therefore, $\text{Var}[Y_{ij}] \geq N d^2 \sigma_X^6 \sigma_W^2$. $\qquad\square$

**Proof of Proposition 2:** If we let $\mathbf{X}_{ij} = a$ be a degenerate R.V. as in worst case equation 3, then $\text{Var}[(\mathbf{XW})_{pq}] = \sigma_W^2 a^2 d$ by C.L.T and properties of variance. In all other cases, from $X_{ij} \in [-a, a]$ follows that $\sigma_{X_{ij}}^2 \leq a^2$ by Popoviciu's inequality (Popoviciu, 1935). Then $\text{Var}[X_{pj}W_{jq}] = \sigma_{X_{pj}}^2 \sigma_{W_{jq}}^2 \leq a^2 \sigma_W^2$, and $\text{Var}[(\mathbf{XW})_{pq}] = \sum_{j=1}^{d} \text{Var}[X_{pj}W_{jq}] \leq \sigma_W^2 a^2 d$ even if some $X_{pj}$ is dependent with some $X_{pj'}$, because $Cov[\sigma_{X_{pj}}^2 \sigma_{W_{jq}}^2; \sigma_{X_{pj'}}^2 \sigma_{W_{j'q}}^2] = 0$ for $j \neq j'$. $\qquad\square$

