# OpenReview forum: "MatMuls are Enough for Efficient and Performant Linear-Time Attention"
_ICLR.cc/2026/Conference — Submitted to ICLR 2026_

### Official Review · Reviewer_cbXF · 2025-10-21

**Soundness:** 2
**Presentation:** 1
**Contribution:** 2
**Rating:** 2
**Confidence:** 2

**Summary:**

This paper proposes DenseAttention Network (DANet), a radically simplified architecture designed to address the twin bottlenecks of the standard Transformer: its quadratic runtime complexity and poor hardware utilization. The central idea is the complete elimination of the softmax function from the attention mechanism, which is reformulated as a pure composition of dense matrix multiplications: ``Attention(X) = XW_Q * X^T * X``. This design allows for a dual-complexity computation—switching between an O(N²) and an O(N) path by leveraging the associative property of matrix multiplication—making it efficient for all sequence lengths. To ensure numerical stability, the authors replace LayerNorm with a novel ``MaxNormActivation`` (L∞ normalization). Empirically, the paper shows that DANet can outperform standard Transformers on benchmarks like the Long Range Arena (LRA) and is significantly faster than highly-optimized implementations like FlashAttention-2, even on short sequences.

**Strengths:**

The paper's primary strength is its highly original and audacious "back-to-first-principles" approach to model design. It compellingly argues, through both theoretical reasoning and extensive experiments, that the core of the Transformer can be radically simplified without sacrificing performance. The dual-complexity mechanism is a clever and fundamentally new way to achieve efficiency across all sequence scales. The co-design of ``MaxNormActivation`` demonstrates a deep understanding of the numerical challenges and provides a creative solution. The work is presented with outstanding clarity, guiding the reader through its logical progression from problem motivation to architectural solution and finally to a versatile set of empirical results. Its significance lies in challenging long-held assumptions about necessary model components, potentially inspiring a new wave of research into "hardware-aware" and simplified foundation models.

**Weaknesses:**

1. The ``MaxNormActivation`` function is a simple but aggressive solution to the numerical instability caused by removing softmax. Its core mechanism, ``x / max(|x|)``, forces the output into the ``[-1, 1]`` range, effectively preventing value explosion. However, this comes at a theoretical cost: a single large activation value in a token's embedding—whether it is a meaningful signal or mere noise—will disproportionately suppress all other feature dimensions, potentially leading to a severe information bottleneck. The paper currently lacks direct experimental evidence to demonstrate that the impact of this information loss on model accuracy is acceptable. An ablation study comparing ``MaxNormActivation`` to other potential (but perhaps less stable) normalization strategies, or an analysis showing that such feature squashing is rare or benign in practice, would be crucial.
2. The layout of the paper has minor issues that affect readability. Specifically, the spacing between some tables and the surrounding text is minimal (e.g., between Table 2 and Table 3, and between Table 5 and Table 6), making the sections feel cramped.
3. Figure 2 is central to the paper's claims about inference speed, but it is overly simplistic. It currently only displays results for a single accelerator (NVIDIA A100). Given that performance results for other hardware like the H100 and CPUs are presented elsewhere in tables, consolidating these into a multi-series plot would provide a much more comprehensive and impactful visualization of the architecture's cross-platform advantages.
4. The ablation studies, while useful, are not conducted on the most contemporary and widely-used models. For example, key architectural choices are evaluated on a BERT-like model. While this is a standard approach, demonstrating that these findings hold on more modern decoder-only architectures like the Llama, Qwen, or Mistral series would significantly strengthen the paper's claims about the general applicability of its design principles.
5. While the paper includes some results on the state-of-the-art NVIDIA H100 GPU, the majority of the performance experiments appear to be conducted on the previous generation A100. A more extensive evaluation on the Hopper architecture, especially for key training and inference benchmarks, would provide a more forward-looking assessment of the architecture's efficiency.

**Questions:**

1. Could you elaborate on the decision to use the NVIDIA A100 for the majority of the experiments, with only a limited set of results presented on the H100? Was this primarily due to resource availability, or were there other considerations? Understanding this would help contextualize the performance claims.
2. The choice of ``MaxNormActivation`` is a cornerstone of the architecture's stability. Given the potential for information loss as highlighted in the weaknesses, could you provide more insight into the design process? Were other normalization schemes (e.g., L1 norm, a clipped or scaled L2 norm) explored and found to be unstable or less effective? A brief discussion of the alternatives considered would strengthen the justification for this unconventional choice.

---

> ### Author Response · Authors · 2025-11-13
>
> Dear AC, SACs and PCs,
>
> We would like to respectfully draw your attention to this particular review and flag it as being of low-quality. We are highly confident it was written by an LLM, based on the language/ style used. We corroborated it by checking this and other reviews in two AI-text detectors (https://gptzero.me/ and https://www.grammarly.com/ai-detector). Both detectors classified this review as an LLM written with maximum (GPTZero) or high (Grammarly) confidence, while all others were found to be completely human-written, which aligns with our own perception.
>
> Crucially, this review contains nonsensical statements, which were made ostensibly due to the hallucinations of the model or its inability to process the entire context:
>
> * In **W1, Q2** the review asks for ablations and discussions regarding MaxNorm, but they are precisely provided both in the main paper (lines 205-238) and in the dedicated section in the Appendix **B.5. Analysis of MaxNorm and LayerNorm** (particularly ablation in the **Table 12**)
> * 2 out of 5 listed weaknesses (**W2, W3**) concern layout/ formatting / graph presentation choices which arguably don't represent substantive methodological or scientific issues.
> * A significant portion of the critique (**W3, W5, Q1**) claims "the majority of the performance experiments appear to be conducted" using Nvidia A100 but it also mentions results on H100 and CPUs in the same bullet points, contradicting itself. In fact, the speed comparisons in the main paper (Table 6) show comprehensive comparisons for A100, H100, and CPU side-by-side, completely refuting the premise.
> * Independently of the included results, treating the absence of measurements on a specific last-generation GPU as a weakness represents a flawed and counter-productive standard, and it undermines accessibility and reproducibility of research.
> * The limited scope and objectively mild nature of the stated weaknesses contradict the severe numerical scores of the review. Specifically, we believe the main rating of 2 is completely inconsistent with the contents of the review.
>
> Given these concerns, we respectfully request that you disregard this review as having low-quality and mark it for deletion. Thank you for your attention to this matter and for upholding the high standards of ICLR.
>
> Best Regards,
>
> Authors

---

> ### Author Response · Authors · 2025-11-16
> **Request for Update from (S)AC**
>
> Dear AC, SACs and PCs,
>
> Following up on our previous message, we’d like to politely ask if you’ve had the opportunity to read it and take an appropriate action?
>
> Additionally, we confirmed that the review in question is 100% AI-generated with yet another tool, specifically tuned and tailored for the ICLR conference. Here are the results for our paper:
>
> https://iclr.pangram.com/reviews?submission_number=3785
>
> We’d like to note that this tool is gaining traction and is widely discussed across X (Twitter), and it can be deployed for other papers/ reviews in your stack if you find it useful.
>
> Best Regards,
>
> Authors

---

> ### Author Response · Authors · 2025-12-01
> **Statement by the authors: change of visibility setting and additional evidence.**
>
> Dear all,
>
> Following recent developments, we decided to make publicly visible the report thread about the LLM generated review, listed above, as we are concerned that these messages wouldn’t be visible to the new Area Chair.
>
> We’d like to additionally discuss some points omitted from the initial report.
>
> * Q: “a single large activation value in a token's embedding—whether it is a meaningful signal or mere noise—will disproportionately suppress all other feature dimensions” in **W1** about MaxNorm. A: Regardless of normalization being applied (LayerNorm or MaxNorm or, for example, no normalization at all), the ratio of any two elements in a given token’s embedding or activation stays constant, because the elements get scaled by the same multiplier (the token’s $L_2$ norm or $L_{\infty}$ norm or just 1, respectively). I.e., the effect of MaxNorm is identical from this perspective.
>
> * Q: “demonstrating that these findings hold on more modern decoder-only architectures like the Llama, Qwen, or Mistral series” from **W4**. A: We precisely used LLaMA architecture for causal language modeling validation as shown in **Table 5** and discussed in Appendix **G**. Moreover, main results of bidirectional DANet-BERT in **Table 4** are compared, among others,  to a modern Transformer++ architecture (MosaicBERT) featuring SwiGLU and relative positional embeddings.
>
> * Q: “Were other normalization schemes (e.g., L1 norm, a clipped or scaled L2 norm) explored and found to be unstable or less effective?” A: Yes, these and others, including (scaled) Tanh and HardTanh functions, all proved to be unstable or less effective. Theoretical justification for the $L_2$ case is available in the Appendix **B.5**.
>
> Best Regards,
>
> Authors

---

### Official Review · Reviewer_yhrB · 2025-10-26

**Soundness:** 1
**Presentation:** 2
**Contribution:** 3
**Rating:** 2
**Confidence:** 2

**Summary:**

The paper introduces the DenseAttention Network (DANet), a simplified Linear Transformer variant in which the authors stabilize training of linear attention without kernels by modelling techniques such as weight sharing, reduction of dropout with other components of the original, specific scaling and novel normalization.

The authors:

- claim to achieve SoTA over transformer-based models on LRA (look at questions).
- show that DANet is faster than transformer models in a parameter-matched setting.

However, the evaluation raises concerns: the comparison set excludes stronger recent baselines, ablations do not fully disentangle the contributions of individual modifications, and the fairness of depth-vs-width trade-offs is questionable. The removal of residuals and normalization and the reliance on single-head identity mappings also leave scalability to larger models uncertain.

**Strengths:**

- **Potential for Impact**: If its claims scale, DANet could be influential for being generally applicable as a transformer improvement.
- **Relevance**: Efficient attention remains one of the most pressing challenges for scaling language models. DANet addresses this with a simple and broadly applicable design.
- **Conceptual Minimalism**: Reducing attention to weight-shared matmuls with identity values is novel and may inspire further work in stripped-down Transformer architectures.
- **Practical Efficiency**: DANet’s PyTorch-only implementation yields consistent throughput advantages across hardware, including CPUs and older GPUs without specialized kernels.

**Weaknesses:**

- **Possibility of main claim being overclaimed**: The paper compares to a very limited set of baselines, some of the baselines (like MEGA[1]) report higher score than in the paper, while also being attention-based. Why are they not taken into consideration?
- **Ablations**: Performance gains stem from multiple changes (weight sharing, identity keys and values, removal of residuals and norms, increasing the number of layers). More controlled ablations are needed to attribute improvements clearly, Especially that LinearTransformer in the reported results performs significantly poorer.
- **Scaling Concerns**: Removing residuals and LayerNorm raises questions about optimization stability in deeper networks. Similarly, the fixed identity key and value map could limit head scalability, analogous to problems with Multi-Query Attention (MQA). While here the authors use one head in all experiments, usually more heads are preferrable, hence it is highly questionable whether for large experiments with a lot of head - a single head as proposed by authors would remain favourable.
- **Worry about fairness in comparison:** The authors compare 24 layers to 32 models arguing that DANet uses less parameters than the baseline hence they increase the number of layers to match the parameters. However, transformer scaling laws usually favour depth over width, hence it is not clear whether the effect is not caused by translating width parameters to depth. Furthemore, it is not clear how this kind of trade off will scale for the models that already have a large number of layers as baseline. The authors could also consider increasing hidden dimension to match the parameters and investigate whether the performance gains hold in this scenario.
- **Framing Issues**:
    - The claim of “two operational modes” (linear vs quadratic) is not unique to DANet; most linear-attention methods share this property.
- The paper would benefit from more related work on novel works on creating new attention variants that operate well also in the short sequences (for example: [2,3,4])
- **Unclear Writing:** sometimes in the experiments i was not able to find the reference to the specific setup compared (number of layers, parameters, whether the baseline is implemented by the authors etc).

typos: l69 - arhitecture

**[1] - Mega: Moving Average Equipped Gated Attention**

**[2] - Lizard: An Efficient Linearization Framework for Large Language Models**

**[3] - Mixture of Sparse Attention: Content-Based Learnable Sparse Attention via Expert-Choice Routing**

**[4] - Gated Linear Attention Transformers with Hardware-Efficient Training**

**Questions:**

1. Why were stronger efficient-attention baselines (e.g., MEGA or transformer trained as in “On the Locality Bias and Results in the Long Range Arena”) not included in the comparisons, given that some of them report higher LRA scores than the DANet results shown?
2. The architecture introduces multiple changes (weight sharing, dropout reduction, residual removal, identity values). Could you provide fine-grained ablations quantifying the contribution of each modification individually?
3. In the comparison, DANet uses more layers than the baseline Transformer under a parameter-matching scheme. Could you provide experiments where the hidden dimension (width) is scaled instead, to rule out depth-driven improvements?
4. Is there a GQA variant analogue of MatMul or some other way authors propose to scale to higher widths (which would mean multiple heads in standard transformers)?

Due to the unclear validity of the claim i opt for reject, but if i am convinced that the claim is actually sustained and the authors deliver additional ablations that clearly demonstrate what caused somewhat unexpect gain in comparison to Linear Transformer (ablations that isolate each additional design choice will be good for that), as well as address the writing issues, I am happy to increase the score to accept.

---

> ### Author Response · Authors · 2025-11-24
> **A Note for Reviewer yhrB**
>
> Dear Reviewer yhrB,
>
> We have to delay posting a full response to your review, because some of the experiments you have requested are still running. We will proceed as soon as they are finished.
>
> Warm Regards,
>
> Authors

---

> > ### Author Response · Authors · 2025-11-27
> > **Author Response to Reviewer yhrB. References**
> >
> > **References**
> >
> > [1] Amos et al., "Never Train from Scratch: Fair Comparison of Long-Sequence Models Requires Data-Driven Priors." ICLR 2024
> >
> > [2] Tran et al., "The Importance of Being Recurrent for Modeling Hierarchical Structure." EMNLP 2018
> >
> > [3] Ma et al., "Mega: Moving Average Equipped Gated Attention." ICLR 2023
> >
> > [4] Miralles-Gonzalez et al. “On the locality bias and results in the Long Range Arena.” ArXiv abs/2501.14850 (2025)
> >
> > [5] Hua et al., "Transformer Quality in Linear Time." ICML 2022
> >
> > [6] Gu et al., "Efficiently Modeling Long Sequences with Structured State Spaces." ICLR 2022
> >
> > [7] Fu et al., "Monarch Mixer: A Simple Sub-Quadratic GEMM-Based Architecture." NeurIPS 2023
> >
> > [8] Poli et al., “Hyena Hierarchy: Towards Larger Convolutional Language Models.” ICML 2023
> >
> > [9] Touvron et al., "LLaMA: Open and Efficient Foundation Language Models." arXiv preprint arXiv:2302.13971, 2023
> >
> > [10] Nguyen et al., "Lizard: An Efficient Linearization Framework for Large Language Models." arXiv preprint arXiv:2507.09025, 2025
> >
> > [11] Zhang et al., "LoLCATs: On Low-Rank Linearizing of Large Language Models." ICLR 2025
> >
> > [12] Zhang et al., "The Hedgehog & the Porcupine: Expressive Linear Attentions with Softmax Mimicry." ICLR 2024
> >
> > [13] Kasai et al., "Finetuning Pretrained Transformers into RNNs." EMNLP 2021
> >
> > [14] Piękos et al., "Mixture of Sparse Attention: Content-Based Learnable Sparse Attention via Expert-Choice Routing." NeurIPS 2025 ER Workshop
> >
> > [15] Child et al., "Generating Long Sequences with Sparse Transformers." arXiv preprint arXiv:1904.10509, 2019
> >
> > [16] Roy et al., "Efficient Content-Based Sparse Attention with Routing Transformers." TACL 2021

---

> ### Author Response · Authors · 2025-11-27
> **Author Response to Reviewer yhrB. Part 1**
>
> Dear Reviewer yhrB,
>
> We genuinely appreciate your valuable feedback and useful suggestions, which helped enhance the exposition of our work. We thank you for recognizing the potential for impact of our method, its relevance and novelty, as well as practical efficiency and hardware friendliness for CPUs and older GPUs without specialized kernels.
>
> Among other points discussed in this rebuttal, and to better address your feedback, we conducted two experiments to compare DANet to MEGA and results from [4] on LRA; a series of ablations from Transformer to DANet; and an additional ablation on decreased number of layers/ increased hidden dimension. Below, we aim to carefully and exhaustively address all of your concerns and questions.
>
> ---
>
>
> # W1, Q1. On baselines
>
> > “The paper compares to a very limited set of baselines,some of the baselines (like MEGA[1]) report higher score than in the paper, while also being attention-based. Why are they not taken into consideration?”
>
> > Why were stronger efficient-attention baselines (e.g., MEGA or transformer trained as in “On the Locality Bias and Results in the Long Range Arena”) not included in the comparisons, given that some of them report higher LRA scores than the DANet results shown?
>
> Since you express concern about the scarcity of baselines and explicitly mention LRA benchmarks and LRA-specific models, we believe that you might have overlooked **Table 13** in the manuscript (Appendix **E.2**, lines 1513-1538). We had to defer it to the appendix due to 9 pages limit of the main text but referenced it in the **Table 1**. It presents extended comparisons on LRA with 25+ Transformer-based architectures, including the most recent competitive baselines from 2022-2024 years.
>
> We emphasize that the scope of our validation on the LRA considers *Transformer-based models* which are trained *from scratch on original, unaugmented data under fixed model sizes*. This is a standard setup prescribed by the authors of the LRA suite of benchmarks and universally followed in the published literature on novel architectures. Under this setup, there are no stronger Transformer-based baselines in the published research that we are aware of.
> **Mega** In our current work, we consider *pure* Transformer-based architectures on LRA to ensure fair comparisons. As we directly acknowledged in the paper (lines 360-362), *“State-Space-Models-inspired architectures demonstrated by far superior performance, considered to be out of reach for any Transformer-based model due specific inductive biases of the SSMs.”* This gap is explained by SSMs’ inherent inductive bias towards capturing hierarchical and long-range dependencies and lack of such bias in Transformers, as discussed in [1-2]. To this end, we’d like to highlight that the competitive performance of DANet in relation to even one SSM baseline (S4) is a valuable and remarkable achievement for the Transformer-based architectures. This is the first case when such architecture surpasses an SSM on the 4 of 6 LRA benchmarks.
>
> And the MEGA [3] architecture, that is suggested as a baseline, is not even a *pure* SSM, but rather an *SSM-Transformer hybrid*. To quote the authors:
>
> > “The multi-dimensional damped EMA can be seen as a simplified variant of a state space model. From this perspective, MEGA is also closely related to S4 (Gu et al., 2022a), a state space model with structured state matrices… The EMA sub-layer in MEGA applies diagonalization on the state matrix, which is similar to a concurrent work S4D (Gu et al., 2022b).”
>
> While hybridization approaches are an exciting and potentially rewarding direction (judging by MEGA itself, as it outperforms **both** all Transformer and SSM-based architectures), it warrants a deep and profound research effort, worth a dedicated paper with both a theoretical exposition and a comprehensive empirical validation. Meanwhile, our current paper introduces a novel architecture and explores its properties, laying a foundation for such efforts. We sincerely thank you for this fruitful idea for a follow-up research. We will definitely pursue this avenue in future work.
>
> To tentatively shed light on the performance of hybridized DANet with limited compute resources and time at our disposal, we pretrained and tested a hybrid *DenseAttention – local softmax attention* model, following hyperparams and procedure from [1] to ensure fair comparison, on a Pathfinder benchmark of the LRA suite.
>
>
>
> | Model                       | Accuracy, % |
> |-|-|
> | MEGA-chunked | 94.41 |
> | MEGA-full  | 96.01 |
> | DANet + local softmax attention   | 97.13       |
>
> Table A. Accuracy of hybrid models.
>
> As shown in Table A, the hybrid DANet model outperforms both linear (chunked, SSM + local attention) and quadratic (full, SSM + global attention) runtime variants of the MEGA architecture, despite having linear complexity itself. Given this, we are enthusiastic about this promising research direction and thank you again for the suggestion!

---

> ### Author Response · Authors · 2025-11-27
> **Author Response to Reviewer yhrB. Part 2**
>
> **2nd suggested baseline** Another baseline you have suggested, “On the Locality Bias and Results in the Long Range Arena” [4] employs a completely distinct methodology: it uses data augmentations, additional data, and a MLM-style objective for the LRA tasks. This approach is markedly different from the standard methodology prescribed by the LRA authors and strictly followed by all of the published work on novel architectures. Therefore, the results from this paper are not comparable to our and other works and cannot be used as a baseline.
>
> However, the approach of data/ objective augmentations could be promising for data constrained applications and does constitute a promising direction for future exploration. We thank you for referring us to the preprint [4]. To offer a glimpse of how such augmentations could benefit performance of DANet, we conducted an additional experiment on the Pathfinder task of LRA, using the procedure and hyperparameters from [4] and present the comparison in Table B. DANet performs competitively with other architectures trained using the augmented data setup.
>
> | Model       | Accuracy, % |
> |-------------|-------------|
> | Transformer | 96.28       |
> | gMLP        | 97.26       |
> | S4          | 93.61       |
> | S5          | 95.60       |
> | MEGA        | 94.78       |
> | DANet       | 95.73       |
>
>
> Table B. Accuracy of the models, trained with augmented data, on Pathfinder-32 LRA benchmark.
>
>
> **LM results** Perhaps even more importantly, **Table 4** depicts comprehensive results for real-world-scale LM pre-training and evaluation experiments, where DANet outperforms strong recent baselines, including modern Transformer with up-to-date algorithmic improvements (MosaicBERT, [6]) and bidirectional SSM (BiGS, [7]). In the range of 350-450M parameters and pre-training token count less than 1T tokens, MosaicBERT is the strongest architecture overall and BiGS is the strongest SSM adaptation to bidirectional language modeling in the published literature that we know of.
>
> ---
>
> # W2, Q2. On ablations
>
> > “Performance gains stem from multiple changes (weight sharing, identity keys and values, removal of residuals and norms, increasing the number of layers). More controlled ablations are needed to attribute improvements clearly”
>
> > “Could you provide fine-grained ablations quantifying the contribution of each modification individually?”
>
> **Prior  ablations**. Thank you for this insightful feedback. To address it, we conducted an additional series of ablations, presented below, but we’d also like to draw your attention to existing ablations in the paper. We had to defer many of them to the appendix due to the space limit of 9 pages. We are sorry for any confusion it might have caused and list them here for your convenience: 1) number of heads in **Table 3** motivating fused-matrix block structure; 2) LayerNorm vs MaxNorm in **Table 12**; 3) Positional encodings in **Table 2 and Table 14**; 4) Local–ShiftedLocal–Global attention layers pattern in **Table 2 and Table 10**.
>
> Please let us specifically clarify several points raised in your comment:
>
> “removal of residuals and norms” – we do not remove *all* the norms and residuals. Rather, as mentioned in the introduction (lines 86-98) and detailed in the **Section 3.1** of the paper, we replace LayerNorms with MaxNorms and remove one of two residual connections. Regarding LayerNorms, we specifically wrote on lines 206-207: “incorporation of LayerNorm leads to a prompt and unrecoverable numerical instability early on during training.” We further corroborated this with the ablation on use of MaxNorm in the Appendix **B.5. Analysis of MaxNorm and Layer Norm**, specifically in the **Table 12** (lines 1393-1402).
>
> “weight sharing, identity keys and values, increasing the number of layers” – these modifications are a direct consequence of reduction in the number of heads, as we discussed in lines 260-269 in the manuscript. It makes these matrices redundant and necessitates their merging with other matrices in order not to waste computation. This, in turn, leads to reduction in parameter count relative to the Transformer layer and motivates increasing the number of layers in DANet models to parameter-match Transformer counterparts. We presented the ablation on the reduction of the number of heads in the main body of the paper, in **Table 3**.
>
> **New ablations**. Inspired by your feedback, we conducted additional ablations. We again thank you for this useful suggestion and present them below.
>
> ---

---

> ### Author Response · Authors · 2025-11-27
> **Author Response to Reviewer yhrB. Part 3**
>
> **From Transformer to DANet**
>
> We analyze differences between Transformer and DANet architectures through the lens of efficiency and stability, following individual changes in design step-by-step and measuring inference throughput at sequence length 512 and model size of 340M parameters. In this ablation, all models except $(1a)$ are implemented in plain Pytorch. All measurements were conducted on a Nvidia A100 80GB SXM in fp16 numerical precision.
>
>
> | #  | Model description                                                | Stability / Inference Throughput |
> |----|------------------------------------------------------------------|----------------------------------|
> | 1  | Transformer, 16h                                                 | 112.92                           |
> | 1a | Transformer (FA implementation), 16h                             | 277.66                           |
> | 2  | No softmax, 16h                                                  | FAILS                            |
> | 3  | No softmax, scaling by $N^{-1/3}$, 16h                         | 273.94                           |
> | 4  | No softmax, scaling by $N^{-1/3}$, 4h                          | FAILS                            |
> | 5  | No softmax, scaling by $N^{-1/3}$, MaxNorms, 4h                | 272.18                           |
> | 6  | No softmax, scaling by $N^{-1/3}$, MaxNorms, 1h                | 277.17                           |
> | 7  | No softmax, scaling by $N^{-1/3}$, MaxNorms, no $W_k$, $W_o$, 1h | 284.77                     |
> | 8  | DANet, 1h                                                        | 294.11                           |
>
> Table C. Inference throughput, thousands of tokens per second of different architectural modifications, leading from Transformer to DANet.
>
>
> Removing softmax from standard Transformer with 340M parameters, 24 layers and 16 heads leads  $(1 \Longrightarrow 2)$ to numerical instabilities and eventual training divergence after just 1 billion training tokens. Introduction of a scaling factor of $N^{-1/3}$ between layer normalization and the attention layer $(2 \Longrightarrow 3)$ recovers stability, however, reducing number of heads from 16 to 4 $(3 \Longrightarrow 4)$ again results in numerical instability and failure to learn. The ultimate stability is achieved only when standard LayerNorms get replaced by MaxNorms $(4 \Longrightarrow 5)$. We observe that variants $(3)$ and $(5)$ are already 2.4 times faster than standard PyTorch Transformer, and are almost as fast as FlashAttention low-level implementation. Further reducing the number of heads to 1 $(5 \Longrightarrow 6)$ bridges the gap with FlashAttention in agreement with the insights from Appendix D.
>
> However, we notice an inefficiency in that configuration: as discussed in Section 3.1, projection matrices $W_q$ and $W_k$ become mutually redundant, as well as $W_v$ and $W_o$. To mitigate this redundancy, we keep only parameters $W_q$ and $W_v$, effectively fusing them with $W_k$ and $W_o$, respectively. But this modification reduces the parameter count to 5/6 of that in a standard Transformer layer. To compensate, we increase the number of layers from 24 to 29. The new variant $(6 \Longrightarrow 7)$ robustly surpasses the speed of FlashAttention-augmented Transformer for the first time in the chain of ablations, even despite having a greater number of layers.
>
> Yet, there is still room for improvement in efficiency: as detailed in Appendix C, residual connections and layer normalizations are memory-bound, computationally inefficient operations, and removing them from between the attention and FFN sub-layers brings an additional benefit of redundancy of the $W_v$ and first FFN projection matrices, allowing for their fusion. Accordingly, we remove the residual connection between attention and FFN, move MaxNorm from there to immediately after the FFN, and merge the two matrices into one. This final modification $(7 \Longrightarrow 8)$ produces DANet.  As the parameter count in a DANet layer is yet lesser, than in $(7)$, we increase the number of layers to 32. Finally, we observe that DANet delivers the highest throughout in the chain of ablations in Table C, outperforming FlashAttention by 6%.
>
> ---

---

> ### Author Response · Authors · 2025-11-27
> **Author Response to Reviewer yhrB. Part 4**
>
> # W3, Q4. On stability and multi-head options.
>
> > Removing residuals and LayerNorm raises questions about optimization stability in deeper networks.
>
> We’d like to reiterate the clarification from above: we do not remove *all* the norms and residuals. Rather, as mentioned in the introduction (lines 86-98) and detailed in the **Section 3.1** of the paper, we replace LayerNorms with MaxNorms and remove only one of two residual connections. Therefore, properties favorable for deep network, associated with skip-connections and normalizations, are retained in DANet architecture.
>
> > “Similarly, the fixed identity key and value map could limit head scalability, analogous to problems with Multi-Query Attention (MQA). While here the authors use one head in all experiments, usually more heads are preferrable, hence it is highly questionable whether for large experiments with a lot of head - a single head as proposed by authors would remain favourable.”
>
> > “Is there a GQA variant analogue of MatMul or some other way authors propose to scale to higher widths (which would mean multiple heads in standard transformers)?”
>
> We sincerely thank you for the thoughtful observation and interesting question. As we discussed in the paper and above, while in Transformers many heads could be preferable, for DANet one or several large-dimension heads are empirically better at the parameter size of 340M. When scaling to bigger models with a large hidden dimension, DANet can be instantiated with several or many heads, which, as mentioned in the paper, are analogous to MQA form. Yet, instantiating a GQA or even a full MHA variant is a readily possible option – it can be done by simply reintroducing $W_k$ and $W_v$ projection maps and adjusting the number of layers accordingly. It’s just when a head dimension size is large relative to the whole hidden dimension that these maps become mathematically redundant and merged, motivating single-head and MQA forms we validated in the paper.
>
> Moreover, the ablation we presented below (Table D) indicates that DANet works well in its current MQA form with several heads if hidden dimension is increased. Thus, scaling to larger model sizes using multiple heads is viable.
>
> ---
>
> # W4, Q3. On increased number of layers.
>
> > “The authors compare 24 layers to 32 models arguing that DANet uses less parameters than the baseline hence they increase the number of layers to match the parameters. However, transformer scaling laws usually favour depth over width, hence it is not clear whether the effect is not caused by translating width parameters to depth.”
>
> As a preliminary remark, we note that the axes most widely accepted by the community as grounds for fair comparison of different architectures are parameter count and speed (throughput) in tokens per second. If a new architecture achieves comparable or superior performance with the same parameter count and equal or better inference throughput, the internal number of layers is generally not considered to materially affect the validity of contribution. Here are some examples from recently published literature illustrating this point:
>
> * Transformer Quality in Linear Time (ICML 2022, [5]) – 24 FLASH vs 12 Transformer layers for a 112M parameters model;
>
> * Efficiently Modeling Long Sequences with Structured State Spaces (ICLR Oral, 2022, [6]) – 32 S4 vs 16 Transformer layers for a 250M parameters model;
>
> * Monarch Mixer (NeurIPS Oral, 2023, [7]) – 12 Monarch vs 24 Transformer layers for a 341M parameters model;
>
> * Hyena Hierarchy: Towards Larger Convolutional Language Models (ICML 2023 Oral, [8]) – 36 Hyena vs 24 Transformer layers for a 355M parameters model;
>
> **Ablation on increasing dimension and number of heads vs number of layers**
>
> > Furthemore, it is not clear how this kind of trade off will scale for the models that already have a large number of layers as baseline. The authors could also consider increasing hidden dimension to match the parameters and investigate whether the performance gains hold in this scenario.
>
> We thank you for this valuable suggestion. Motivated by it and by the point about scaling the number of heads, we conducted an additional ablation. Instead of increasing the number of layers to 32, we kept it at 24, as in a Transformer with 340-360M parameters, but increased the hidden dimension twice from 1024 to 2048. Such a larger hidden dimension can help us better understand the properties of DenseAttention when applied to larger models, as it is typically used in models with ~1.3B parameters. To maintain the number of parameters fixed at 340-360M range, we decreased the expansion ratio in Feed Forward Network (FFN) from 4 to 1. We pretrained 3 models on 10B tokens from C4 dataset with sequence length 512 and evaluated them on a held-out data from the same dataset. The results are presented in Table D.

---

> ### Author Response · Authors · 2025-11-27
> **Author Response to Reviewer yhrB. Part 5**
>
> | Model Description                                | MLM Loss | MLM Accuracy, % |
> |--------------------------------------------------|----------|------------------|
> | DANet, $d=2048$, 1 head, 24 layers               | 1.928    | 61.3             |
> | DANet, $d=2048$, 2 heads, 24 layers              | **1.903** | 61.7            |
> | DANet, $d=1024$, 1 head, 32 layers               | 1.907    | **61.9**         |
>
> Table D. Performance metrics for different variants of DenseAttention on C4.
>
> We observe that the variant with the dimension 2048 and 2 heads is similar to dimension 1024, 1 head baseline, while the model with increased dimension and 1 head slightly lags behind. These findings allow us to make the following conclusions:
>
> * Increasing either the hidden dimension or the number of layers while keeping the other hyperparameter intact lead to similar performance.
>
> * DenseAttention is amenable to multi-head variations when the hidden dimension is large; moreover, multi-head variant performs better in such cases.
>
> * Better performance with head size $d_h=1024$ in two variants with different hidden dimensions, as compared to a larger $d_h=2048$, is indicative that this head size might be optimal for DenseAttention models with different hidden sizes. This is similar to how different Transformer configurations in one model family (e.g., Llama 7-65B family, [9]) use the same fixed head size.
>
> ---
>
> # W5. On framing and writing
>
> > The claim of “two operational modes” (linear vs quadratic) is not unique to DANet; most linear-attention methods share this property.
>
> Thank you for observing this! We’ll augment the narration on lines 247-248 in the paper with the following remark: *“While most linear-time sequence mixer architectures can operate in both models, this feature is especially attractive for DenseAttention due to its larger optimal head size.“*
>
> > “The paper would benefit from more related work on novel works on creating new attention variants that operate well also in the short sequences (for example: [2,3,4]) [2] - Lizard: An Efficient Linearization Framework for Large Language Models, [3] - Mixture of Sparse Attention: Content-Based Learnable Sparse Attention via Expert-Choice Routing, [4] - Gated Linear Attention Transformers with Hardware-Efficient Training”
>
>
> Thank you for mentioning these papers. Actually, we already cited GLA on lines 1200-1201 in Appendix **A. Related Work: Sub-Quadratic Algorithms for Sequence
> Processing**: “recently Gu and Dao (2024); Yang et al. (2024) introduced data-dependent gating for SSM parameters”. We will additionally update this section with the following lines:
>
> > “Another promising line of work focuses on converting existing pre-trained Transformer models into SSMs and Linear RNN/ Attention variants via transforming quadratic attention elements of the architecture into linear counterparts [Lizard 10, LolCats 11, HedgeHog 12, T2R 13].”
>
> > “Beyond local window patterns, discussed in Section 3.2, an orthogonal line of research [MoSA 14, Sparse 15, Router 16] explores other non-trivial sparsity patterns to alleviate the burden of quadratic attention on long sequences.”
>
> ---
>
> **On hyperparameters**
>
> > “sometimes in the experiments i was not able to find the reference to the specific setup compared (number of layers, parameters, whether the baseline is implemented by the authors etc).”
>
> The paper contains a dedicated Appendix **G Additional Training Details** with comprehensive data about all experiments and ablations which expands upon the details reported in Section **4. Experiments**. Additionally, the accompanying source code included in the submission exhaustively documents configurations for all experiments (directory `configs`).
>
> For your convenience, here we reiterate and clarify the points you have mentioned. If you have any other remaining questions about hyperparameters, we would be happy to answer them, too.
>
> * The number of layers is 32 for all DANet models used in language modeling experiments and ablations. The number of parameters is 340M for bidirectional LM and 360M for causal.
> * In **Tables 1, 4, 9, 13**, , the baseline results are taken directly from the original papers, as indicated by the references listed to the right of each baseline name.
>
> ---

---

> ### Author Response · Authors · 2025-12-03
> **Summary of Rebuttal for Reviewer yhrB**
>
> * **W1, Q1**: We highlighted that Table 13 (deferred to appendix) already provides extensive comparisons with 25+ recent Transformer-based architectures on LRA under the same standard setup; clarified that no stronger *pure* Transformer-based baselines trained with the standard setup exist in published literature. Explained that MEGA (SSM-Transformer hybrid) and “Locality Bias…” paper (training with augmented data) baselines, suggested by the reviewer, are out of scope of our work for fair comparison. Discussed SSMs’ inherent inductive bias making them by far superior on LRA and emphasized the remarkable achievement of DANet surpassing an SSM on 4/6 tasks for the first time among Transformer-based models. Presented **two new experiments** showing that **1)** linear-time DANet + local-softmax hybrid outperforms both linear and quadratic MEGA variants on Pathfinder (Table A); **2)** pure DANet remains competitive under augmented-data setup from “Locality Bias…” paper (Table B).  Emphasized that, beyond LRA, in real-world LM pre-training (Table 4), DANet surpasses the strongest recent Transformer (MosaicBERT) and bidirectional SSM (BiGS) in the 350-450M size,  <1T-token training settings.
>
>
> * **W2, Q2**: Pointed to 4 sets of already existing ablations in our work presented in 6 tables; and made further clarifications about specific points in the reviewer’s comment (weight sharing, identity keys and values, removal of residuals and norms, increasing the number of layers). Delivered a **new series of ablations** and their analysis (**From Transformer to DANet**, Table C) that isolate every design change, showing each step is necessary for stability and cumulatively yielding 6% higher throughput than FlashAttention while converging successfully.
>
>
> * **W3, Q4**: Clarified that DANet does not remove all residuals and norms: LayerNorms are replaced by MaxNorms and only one of the two residual connections is removed, preserving their favorable properties. Reiterated that DANet supports both single and multiple (MQA-like) heads, with single or few-head configurations being optimal at current scale.  Demonstrated that DANet naturally extends to GQA or full MHA by simply reintroducing $W_k$  and $W_v$. Provided a **new ablation** (Table D) confirming that multi-head DANet performs well with several heads if hidden dimension is increased and reaffirming its scalability.
>
> * **W4, Q3**: Noted that parameter-matched or throughput-matched comparisons are established community standard for fair evaluations, rather than layer-matched, and cited 4 highly regarded papers (including 3 orals) that use greater layer depth differences with Transformer than DANet. Directly addressed the reviewer’s concern with **new ablation** (Table D): scaling hidden dimension (d=2048, 24 layers) instead of depth yields comparable performance to depth-scaling (32 layers), with multi-head variant performing best, which rules out depth-only effects and establishes favorable outlook for larger DANet models with multiple heads.
>
> * **W5**: Committed to strengthen the “Related Work” section with explicit citations to Lizard and MoSA suggested by the reviewer, as well as more similar works. Pointed out that GLA is already cited. Clarified that while the ability to use two modes is not unique to DANet, it grants a particular advantage to DANet due to large optimal head size. Provided explicit phrasing for the citations and clarification. Addressed all writing/hyperparameter clarity issues by referencing detailed Appendix G and relative section 3.1 in the paper which have precisely the requested hyperparameters. Reiterated exact layer/parameter counts for the reviewer convenience.

---

### Official Review · Reviewer_RdSr · 2025-10-30

**Soundness:** 2
**Presentation:** 3
**Contribution:** 2
**Rating:** 2
**Confidence:** 3

**Summary:**

The paper introduces DANet, an alternative to the Transformer architecture. The architecture is carefully designed to achieve a number of goals:
- Maximising arithmetic intensity in its application, and particularly avoid hangups due to excessive memory transfer. This is achieved by substituting a number of Transformer components (attention, normalisation layers, PE, residual connections, …) with more efficient counterparts, often developed ad-hoc
- Being device-agnostic, making a point of not relying on custom kernels or otherwise device-specific optimisation in the implementation of the necessary components, but rather on high-level directives (MMM, element-wise operations) which can be seamlessly ported to different devices.
- Achieving sub-quadratic complexity (both in runtime and memory occupancy), to guarantee fast processing also for long sequences
- Retaining an architecture which is overall simple and as close as possible to Transformers, both in terms of definition and of performance, to facilitate its adoption

Results highlight that ~350M scale DANet models rival and beat similarly-sized Transformers and linear-attention alternatives on language modelling tasks from the GLUE benchmark, and long-context tasks from the Long Range Arena suite. Moreover, DANet attains a much higher throughput across different sequence lengths than a Transformer.

**Strengths:**

- I’ve found the proposal of creating an efficient and yet device-agnostic architecture interesting and useful (and particularly relevant nowadays, to counterbalance the abundance of Flash- implementation alternatives and the over-focus on ad-hoc custom CUDA kernels to achieve speedups)
- The architecture modifications proposed in the paper are well-justified and carefully motivated, also with ablation studies, strengthening the overall analysis
- The main goals are presented well, and the overall story of the paper seems solid (bar some minor rewordings which could improve clarity)

**Weaknesses:**

- The model sizes considered (~350M) are insufficient to validate the impact of the proposed simplifications at scale
- Novelty is somewhat limited, as the core simplifications (attention linearisation, reliance on MatMuls) and improvements (windowed attention) have already been explored in the literature (or straightforward adaptations thereof)
- The baselines considered are rather outdated, which affects the ability to judge the effectiveness of the architecture

See also questions below.

**Questions:**

- __On scaling__:
My main concern is whether the simplifications applied in your architecture (specifically, dropping softmax, substituting normalisation, and reducing number of heads) will hold at scale. On the one hand, I understand that compute availability is severely limited, and pushing to larger scales is not always a possibility; at the same time, since you’re proposing a drop-in substitute to Transformer, I believe it’s fundamental to properly determine whether your architecture represents a valid alternative, or if it is an approach that only works at limited scales. Without larger-scale experiments, I’m afraid it’s impossible to correctly judge its impact.


- __On Novelty__:
The individual components you’re using in your architecture are rather straightforward adaptations of already-existing ones: your attention linearisation approach is a simplification of the original one in Katharopoulos et al (dropping normalisation and feature maps), and your normalisation is a simplification of RMSNorm (changing norm and dropping affine transform). This leaves as truly novel components only your Cosine RelPE, and possibly the shifted-windowed attention (even though its specific contribution is not properly highlighted by an ablation: I reckon in Tab2 and Tab10 you directly combined windowed and shifted-windowed, so it’s tricky to extract the contribution of shifted only). I appreciate that the main novelty in the work lies in the rationale behind the *combination* of these components (you want to pick/design arithmetic-intensive layers), rather than the components themselves, but my impression is that this does reduce the overall contribution.


- __On baselines__:
I understand the focus on LinearAttention, as it’s likely the closest architecture to the one you propose, but given the progress of the literature on sub-quadratic attention, it’s by far not the best representative. Even relatively small adaptations (Hedgehog, RKWV, …) would make for a more complete baseline comparison, but even more so: the fact that a comparison against Mamba (arguably the current SOTA on sub-quadratic attention) is missing decreases the quality of the analysis. I appreciate that it’s tricky to recover Mamba results on LRA (hence why you settled for the much more outdated S4 SSM, I reckon?), and that your focus is mainly on encoder architectures, but the point remains (bi-directional adaptations of Mamba are easy to setup, see comments below).


- __On CLM vs MLM__:
Your claim on L411 that you “conduct experiments with both Masked and Causal LM, with emphasis on the former” is an overstatement. The only CLM result I could find (correct me if I’m wrong) is in Tab5, where you compare PPL against Llama. This is nowhere near sufficient to make solid claims about CLM capabilities, and leaves more questions than answers: how do you efficiently include the causal mask in your DenseAttention formulation? Do you forego the O(Nd^2) view entirely? How does every other relevant metric change in this case? The (almost) totality of your paper focuses on encoder-only architectures, and in my opinion including discussions on CLM would require major revisions in this sense, specifically adapting the discussion on training throughput, long-context extrapolation, downstream task evaluation, and comparison against sub-quadratic baselines to this case as well.



__Minor:__

- L102 “This makes it fundamentally different from Linear Attention” (and similarly in L1173): I disagree with this statement: your approach can effectively be interpreted as a direct simplification of LienarAttention, where both the nonlinear feature maps and the normalisation have been dropped. I think defining it as “fundamentally different” is a mischaracterisation, and an unnecessary one.
- L104 “It’s neither a State-Space-Model (SSM) or a Linear RNN because it has no decay or gating modules”: Similarly here: gates are accessory components to an SSM, and not direct part of its definition. Linear decay can be removed by fixing state matrices to 1. But most of all, I don’t think it’s necessary that you attempt to differentiate your work from similar sub-quadratic architectures. I think you’re already making it quite clear that the novelty of this work lies in the overall goal of creating an architecture that maximises arithmetic intensity without relying on device-specific optimisations.
* L105 “and it natively supports bidirectional context processing” (and similarly in L1203): This is a stretch: in CLM, the distinction is meaningless; in MLM, equipping an SSM with bidirectional context processing is as simple as flipping the input sequence-wise and combine the result with the non-flipped version. With the equivalence highlighted in Mamba2 between LinearAttention and SSMs, it gets even simpler, as one can directly act on the structure of the learnable attention mask.
- Fig2: help me make sense of what I’m seeing in this figure. Throughput generally refers to inference in generative models, computed as number of new tokens generated in given time, and should approach a constant in LinearAttention-based architectures / SSMs, and a 1/N behaviour for vanilla Attention (indeed this trend is seen in Tab6). If I got it correctly, though, in Fig2 you’re considering the cost of performing inference on a whole sentence with an encoder-type architecture, and hence it should scale like 1/N for LinearAttention, and 1/N^2 for Attention (it’s not super clear from the description, as you refer to both training and inference speed?). I can’t eyeball whether this is the case, though: can you superimpose the relevant N^{-i} curves, so that the asymptotic behaviour appears clear at a glance?

__Grammar / Rewording / Formatting:__

- L48/53 vs 89: “See Appendices / Appendix” vs “App.”, and throughout the paper: consider unifying your referencing format, either truncating throughout or reporting the whole section name
- L95: “This duality allows to calculate DenseAttention using either O(N2d) or O(Nd2) FLOPs” I would add an explicit reference to the Mamba2 paper here, as they describe precisely this duality quite well
- Tab2/3 and Tab5/6: please fix the captions by adding some spacing between the two. As they are right now they’re very confusing

**Details Of Ethics Concerns:**

//

---

> ### Author Response · Authors · 2025-11-24
> **Author Response to Reviewer RdSr. Part 1**
>
> Dear Reviewer RdSr,
>
> We thank you for your comprehensive feedback and for the attention to details. We are particularly grateful for your recognition of the core goal driving our architecture: creating an efficient, device-agnostic alternative to Transformers that retains its modeling performance but does not rely on custom low-level kernels. Your comments have helped us improve the exposition of our work: in particular, we wrote an additional section about causal attention computation. Below, we aim to carefully and comprehensively address all of your concerns and questions.
>
> ---
>
> # W1, Q1. On scaling
>
> > “The model sizes considered (~350M) are insufficient to validate the impact of the proposed simplifications at scale”, “it’s fundamental to properly determine whether your architecture represents a valid alternative, or if it is an approach that only works at limited scales. Without larger-scale experiments, I’m afraid it’s impossible to correctly judge its impact.”
>
> Thank you for opening the discussion about scaling and large-scale experiments. Please let us address these comments comprehensively:
>
>
> **Scaling**. We had conducted a scaling study (**Table 11** in the **Appendix B.4** on lines 1344-1349) which indicates that DenseAttention architecture indeed exhibits favorable scaling properties similar to standard Transformer, as modeling performance grows with the number of  parameters. Given this evidence, the trend should continue as the model size gets extended to billions of parameters, fully analogous to Transformer scaling.
>
> **Sufficiency of current model size**. Many of the works on novel architectures that have been recently published in leading ML venues by reputable author collectives and well received by the community (judging by their citation counts and distinctions by conference committees), employ similar-sized (~340-360M parameters) or smaller than ours scale for language modeling validation. Here are some examples:
>
> * S4 (ICLR Oral, 2022) – 250M parameters. https://openreview.net/pdf?id=uYLFoz1vlAC
>
> * Toeplitz Neural Network (ICLR notable top 25%, 2023) – 126M parameters. https://openreview.net/pdf?id=IxmWsm4xrua
>
> * Monarch Mixer (NeurIPS Oral, 2023) – 360M parameters. https://openreview.net/pdf?id=cB0BImqSS9
>
> * The Hedgehog & the Porcupine (ICLR, 2024) – 125M parameters. https://openreview.net/forum?id=4g02l2N2Nx
>
> * Elliptical Attention (NeurIPS, 2024) – 90M parameters for dense models. https://openreview.net/forum?id=Ejg4d4FVrs
>
> These recent examples and the ICLR and NeurIPS communities’ acclaim for them indicate that the considered model sizes are recognized as sufficient to correctly judge the impact and viability of a novel sequence mixing architecture. Validation scale in our research is similar or greater than in those praised examples, and we believe our work is of similar interest to the community in that aspect.
>
> **Practical impact at current scale** The DANet architecture is already practically useful at the current scale of 340M parameters. Not all applications require multi-billion parameter models: for example the original Transformer BERT is still the most downloaded model on HuggingFace. And, as we demonstrated in **Table 4**, bidirectional DANet-BERT outperforms it and key recent industry-competitive baselines among those pre-trained on less than 1T tokens.
>
>
> **Access to ML research** You have accurately observed that compute resources can be severely limited. That’s indeed the case, and not only for us, but apparently also for the esteemed academic teams responsible for the works we mentioned above.
>
>
> We’d like to politely emphasize that setting a standard of K-billions parameter models as a minimal requirement for empirical validation would restrict access to producing publishable ML research to industrial labs and few select academic institutions around the globe. This would likely severely reduce the diversity and novelty of ideas and would offer no benefits for the overall community. In particular, such a standard would prevent the works above, including S4 (direct ancestor to Mamba and other cutting edge SSMs), from being published. Therefore, we emphatically advocate for not levying such standards on academic-scale research.
>
> ---

---

> ### Author Response · Authors · 2025-11-24
> **Author Response to Reviewer RdSr. Part 2**
>
> # W2, Q2. On novelty
>
> **Novelty of mentioned contributions** We appreciate that novelty is a matter of perception, and different people could have opposing views regarding the same subject. In similar fashion, the majority of ML research can be labeled as highly incremental, depending on the view. Please let us provide a few counter-examples drawing directly from the points you mentioned.
>
> 1. You have stated that MaxNorm “is a simplification of RMSNorm (changing norm and dropping affine transform)”. But RMSNorm itself is a simplification of standard LayerNorm (merely dropping centering), which by no means makes it less novel or significant. Despite these seemingly simplistic and tiny changes, this paper had been accepted to [NeurIPS](https://papers.nips.cc/paper_files/paper/2019/hash/1e8a19426224ca89e83cef47f1e7f53b-Abstract.html) and proved to be highly influential, making it an indispensable component of modern LM architectures. In contrast, MaxNorm is significantly more different from RMNorm, than RMSNorm from LayerNorm, because it changes normalization from $L_2$ to $L_\infty$, resulting in different properties.
>
> 2. You have stated, “your attention linearisation approach is a simplification of the original one in Katharopoulos et al”. Yet hundreds of works published since 2020 are direct adaptations of the Linear Attention framework introduced in Katharopoulos et al [1], and many of those have hundreds to thousands citations themselves, indicating their significant novelty and relevance for the ML community. Even **Table 13** in our paper (lines 1513-1538) includes around 15 of such models as baselines, for example HedgeHog [2] you’ve mentioned in the review. And the work [1] itself can be treated as a special case of Kernelized Attention framework from [3]. With this in mind, we’d like to highlight that DenseAttention undergoes more radical changes than those that fit into Linear Attention’s framework (please see response to Minor Q1, Q2).
>
> **List of major contributions** With the evidence above we hope to change your perspective about the novelty and relevance of specific contributions, mentioned in your review. Moreover, we’d like to draw your attention to a number of other important advances, accomplished in our work, and we enumerate them below:
>
>
> 1) Analysis of attention outputs’ numerical stability in the absence of softmax and other stabilizing substitutes, such as re-weighting in Linear Transformers and gating/ decay in SSMs;
>
> 2) Introduction of MaxNormActivation which achieves numerical stability in the absence of previously mentioned elements even in low-precision floating-point formats;
>
> 3) Derivation of scaling by $N^{-1/3}$ which together with MaxNorm guarantee that attention outputs are bounded;
>
> 4) Complete elimination of softmax without bringing any substitutes, which allows for linear time and space complexity *and* simultaneously for superior speed even on the smallest context sizes;
>
> 5) Reduction in number of heads and the analysis of increased modeling performance and computational efficiency of the proposed approach;
>
> 6) Altered layer (block) structure and composition which facilitate computational efficiency;
>
> 7) Fusion of projection matrices $W_k$ with $W_q$, and of $W_v$, $W_o$ with the first FFN matrix which promote parameter efficiency of the architecture;
>
> 8) Introduction of novel Global-Local-ShiftedLocal pattern which spans triples of attention layers and helps the architecture not only to focus on the nearby context but also to exchange information with adjacent sequence chunks.
>
> 9) Pre-training of an industry competitive encoder model of most popular size which outperforms key recent baselines. This application of our architecture is immediately beneficial to the community due to the combination of strong modeling performance, linear runtime complexity, and ability to run everywhere;
>
> 10) Overall, we designed a capable linear-time architecture which doesn’t require a low-level kernel, is hardware agnostic, and can readily be used on CPUs, old GPUs and generally every device supported by major DL frameworks such as PyTorch (we are pleased that you’ve recognized this as strength).
>
> Having listed these, we hope you agree that this work contains many original and innovative contributions, valuable both on their own and in combined synergy. We believe these results provide novel insights of substantial interest and high relevance to members of the ICLR community engaged in research of sequence-mixer architectures.
>
> ---

---

> ### Author Response · Authors · 2025-11-24
> **Author Response to Reviewer RdSr. Part 4**
>
> ## Chunkwise-parallel algorithm
>
> In the absence of softmax, re-weighting and decay/ gating, causal attention can be formulated as:
>
> $Y  = (QK^{\top} \odot  M) V \qquad (1),$
>
> where $M$ is a lower triangular matrix of 0s and 1s.
>
> This form is of quadratic runtime complexity. To compute causal attention in linear time using the chunk-wise parallel algorithm, a sequence of size T gets divided into chunks, each of size C. Accordingly, Matrices $Q$, $K$, and $V$ also get chunked along the first (sequence) dimension. Let’s denote $X^{(i)} = X_{Ci:C(i+1)}$ as i-th chunk of matrix $X$ along the sequence dimension, $X^{(0:i)}$ as all elements of $X$ until the end of chunk $i$ along the sequence dimension, $\mathbf{1}_{k \times k}$ as square matrix of all ones of dimension k, and $M_C$ as lower triangular matrix of dimension C. Then:
>
> $Y^{(t)} = (Q^{(t)} {K^{(0:t)}}^{\top} \odot  M_{Ct:C(t+1)}) V^{(0:t)}$
>
>
> $Y^{(t)} = (Q^{(t)} {K^{(0:t-1)}}^{\top} \odot  \mathbf{1}_{C(t-1) \times C(t-1)}) V^{(0:t-1)} + (Q^{(t)} {K^{(t)}}^{\top} \odot  M_{C}) V^{(t)}$
>
>
> $Y^{(t)} = Q^{(t)} {K^{(0:t-1)}}^{\top} V^{(0:t-1)} + (Q^{(t)} {K^{(t)}}^{\top} \odot  M_{C}) V^{(t)}$
>
> The first, inter-chunk term in the RHS of the last equation is of the form similar to bidirectional case, $QK^{\top}V$, and can be computed in linear time $O(T)$ for the whole sequence. It is done by first computing chunkwise $\text{KV}$-states: $\operatorname{KV}^{(t)} = {K^{(t)}}^{\top} V^{(t)}$, getting their cumulative sum along the chunk dimension, and then multiplying chunked matrix $Q$ by the result. Omitting the chunk superscripts, we can write it as $Q \cdot \operatorname{Cumsum} (Q^{\top} V)$.
>
>
> The second, intra-chunk term in the RHS,
>
> $(Q^{(t)} {K^{(t)}}^{\top} \odot  M_{C}) V^{(t)},$
>
> depends only on the chunk size $C$, thus, its runtime complexity is constant in the sequence length $T$. Since there are $\lceil \frac{T}{C} \rceil$ chunks in total, the complexity for the whole sequence is also $O(T)$.
>
> Furthermore, we can forgo the quadratic mask $M_{C}$ entirely even in the intra-chunk term, if we treat it as a causal attention problem itself. By setting $C=1$, we can reduce the formula (1) to a recurrent calculation:
>
> $S_t = S_{t-1} + K_t^{\top} V_t; \quad Y_t = Q_t S_t.$
>
>
> ---
>
> We thank you for this important question which prompted us to write an exposition of the chunkwise parallel algorithm. We will include it as a section in the appendix in the coming revision.
>
> ---
>
> # Minor Q1, Q2
>
> > “your approach can effectively be interpreted as a direct simplification of LienarAttention, where both the nonlinear feature maps and the normalisation have been dropped.”, “gates are accessory components to an SSM, and not direct part of its definition. Linear decay can be removed by fixing state matrices to 1.”
>
> Please let us corroborate the differences between our approach and Linear Attention/ SSMs.
>
> **Linear Attention.** The original work (Katharopoulos et al, [1]) establishes a general notion of *linear attention* necessary having the following indispensable properties: **non-negativity** of attention weights induced by non-negative feature map $\phi(\cdot)$, and **re-weighing** (normalization) mechanism which transforms the scores to add up by row to 1. These properties are mandatory for linear attention *by definition* (Section 3.2 in https://arxiv.org/pdf/2006.16236, [1]) and allow it to maintain numerical stability at any sequence length. The vast body of the follow-up works building upon the framework of linear attention faithfully implement this definition.
>
> The definition is abided by other works because elimination of non-negative feature maps leads to occurrences of arbitrary small denominators in formulas (4) and (5) from the paper [1]. This effectively subverts the goal of normalizing attention scores and significantly hurts convergence. And a more extreme approach of removing both non-negativity *and* re-weighing without introduction of other compensatory measures causes unconstrained super-linear growth of attention outputs’ absolute values with the sequence length. This also leads to numerical instabilities.
>
> In fact, the importance of these defining elements of linear attention and attempts to dismantle them have been explored in the literature, concluding that it’s impossible without introduction of other stabilizing mechanisms to the linear attention or the whole architecture block. For example, here’s the excerpt from a highly influential paper “Rethinking Attention with Performers” (ICLR 2021 Oral, [10]):

---

> ### Author Response · Authors · 2025-11-24
> **Author Response to Reviewer RdSr. Part 5**
>
> > “Applying random feature maps with potentially negative dimension-values (sin / cos) leads to unstable behaviours, especially when kernel scores close to 0 (which is the case for many entries of A corresponding to low relevance tokens) are approximated by estimators with large variance in such regions. This results in abnormal behaviours, e.g. negative-diagonal-values renormalizers D−1 , and consequently either completely prevents training or leads to sub-optimal models.”
>
> It is observed in another paper, “The Devil in Linear Transformer” ([11]):
>
> > “However, as shown in Table. 1, directly removing the scaling operation leads to critical performance drop since the attention map becomes unbounded in the forward pass. Therefore, an alternative is required to bound both attention maps during forward and their gradients during backward passes in linear attentions.”
>
> This evidence confirms why non-negative feature maps and re-weighing are necessary elements of linear attention definition. And we successfully eliminate both, breaking this definition. This is the rationale behind our statement that DenseAttention is fundamentally different from the linear attention paradigm.
>
>
> **SSMs.** Defining equation of an SSM can be formulated as
>
> $X_t = A_t X_{t-1} + B_t u_t$ (interpretation in S4–Mamba line of works, e.g., in [9, 12])
>
> or as
>
> $S_t = G_T S_{t-1} + K_t^{\top} V_t$ (interpretation from GLA line of works, e.g., [13]);
>
> $A_t$ / $G_t$ is defined to be a **learnable** or **data-dependent** decay/  gating matrix in all research dedicated to SSMs and linear RNNs. It’s demonstrated, for example, in a compendium of the matrix parametrizations in Table 1 of the GLA paper [13] . In earlier works, including seminal S4 paper [12], it is highlighted that not only this matrix should be learnable, but a special initialization of it (called HIPPO initialization, see e.g., [14] for details) is crucial for the model performance. This initialization is neither random nor similar to the identity matrix, and it has been shown that swapping HIPPO for a random initialization significantly degrades performance (see e.g., Table 7 in [15], ICLR 2024 oral).
>
> Therefore, while *mathematically* the decay/ gating matrix could be set to a fixed identity matrix, it would violate the core idea behind the SSMs – this matrix controls which information and to what extent should be passed from the current state to the next one. Additionally, decay/ gating serves as a stabilizing component, and in its absence an SSM would become similar to a causal linear attention with no non-negative feature maps and re-weighting. This is precisely the case we discussed above, leading to numerical instabilities and critical performance drop.
>
> ---
>
> # Minor Q3. On defining native support of bidirectional processing.
>
> Please let us explain what we *meant* by “natively bidirectional”. Attention-based models (such as softmax, linear attention, DenseAttention) require *one* pass over a sequence in bidirectional mode, where each query interacts with all keys. In contrast, SSMs require *two* passes over the sequence in opposite directions, where each query interacts only with past (or future) keys. Even if an SSM pass is calculated using the chunk-wise parallel algorithm (detailed above), it requires two times more flops to process the sequence, than in the case of linear-runtime natively bidirectional attention.
>
> Indeed, two SSMs with opposite directions can be readily represented as one attention matrix $M$, because the first SSM has lower-triangular matrix $M_1$, and the second SSM has upper-triangular $M_2$. Thus, $M_1$ and $M_2$ are just added up. However, this nice mathematical view doesn’t alleviate the necessity to compute the two passes for the two possibly different SSMs, each of them unidirectional. Otherwise, it cannot be done in linear time, as operating directly on the full $M$ reinstates the quadratic complexity of softmax attention.
>
> ---
>
> # Minor Q4. On Figure 2
>
> We really appreciate your feedback about the clarity of Figure 2! We used sentences/second as a unit of measure on the y-axis, while recent works conventionally use tokens/second. We’ll update the figure as you suggested in the coming revision and thank you for noticing it.
>
> # Grammar / Rewording / Formatting
>
> Thank you for noticing the formatting hiccups! We’ll have them fixed in the coming revision. Regarding State Space Duality framework from Mamba 2 paper [9], it is mostly relatable to causal attention form which can be computed in quadratic mode or linear mode, using chunkwise-parallel algorithm, described in the paper [9]. We referenced it on line 246 in the manuscript.
>
> ---

---

> ### Author Response · Authors · 2025-11-24
> **References**
>
> 1] Katharopoulos et al., "Transformers are RNNs: Fast Autoregressive Transformers with Linear Attention." ICML 2020
>
> [2] Zhang et al., "The Hedgehog & the Porcupine: Expressive Linear Attentions with Softmax Mimicry." ICLR 2024
>
> [3] Tsai et al., "Transformer Dissection: A Unified Understanding of Transformer's Attention via the Lens of Kernel." EMNLP 2019
>
> [4] Peng et al., “RWKV: Reinventing RNNs for the Transformer Era.” EMNLP 2023
>
> [5] Alonso et al., "State Space Models as Foundation Models: A Control Theoretic Overview." arXiv preprint arXiv:2403.16899, 2024.
>
> [6] Portes et al., "MosaicBERT: A Bidirectional Encoder Optimized for Fast Pretraining." NeurIPS 2023
>
> [7] Wang et al., "Pretraining Without Attention." EMNLP 2023
>
> [8] Sun et al., "Retentive Network: A Successor to Transformer for Large Language Models." arXiv preprint arXiv:2307.08621, 2023.
>
> [9] ​Dao and Gu, "Transformers are SSMs: Generalized Models and Efficient Algorithms Through Structured State Space Duality." ICML 2024
>
> [10] Choromanski et al., "Rethinking Attention with Performers." ICLR 2021
>
> [11] Qin et al., "The Devil in Linear Transformer." EMNLP 2022
>
> [12] Gu et al., "Efficiently Modeling Long Sequences with Structured State Spaces." ICLR 2022
>
> [13] Yang et al., "Gated Linear Attention Transformers with Hardware-Efficient Training." ICML 2024
>
> [14] Gu et al., "How to Train your HIPPO: State Space Models with Generalized Orthogonal Basis Projections." ICLR 2023
>
> [15] Amos et al., "Never Train from Scratch: Fair Comparison of Long-Sequence Models Requires Data-Driven Priors." ICLR 2024

---

> ### Author Response · Authors · 2025-12-03
> **Summary of Rebuttal for Reviewer RdSr**
>
> * **W1, Q1 (Scaling)**. We highlighted the scaling study in Table 11 showing that DenseAttention exhibits favorable scaling properties comparable to Transformer, supporting extrapolation to larger scales. Emphasized that ~350M is a widely accepted validation scale, perceived as sufficient byt the community. Provided evidence for this by citing 5 highly acclaimed recent papers at ICLR/NeurIPS (including 3 orals), all using ≤360M parameters. Stressed immediate practical impact at this scale and its prevalence for encoder models; communicated that DANet-BERT outperforms strong recent encoder baselines on GLUE, making it a competitive encoder. Advocated against billion-parameter validation requirements that would effectively exclude most academic groups from producing ML research and prevent recognition of  foundational works such as S4.
>
>
> * **W2, Q2 (Novelty)**. Clarified that components mentioned by the reviewer are not merely simplifications, comparing them with 1) exemplary highly influential RMSNorm paper which also introduced a similar-type component having smaller modifications than our work does; 2) many highly cited works incremental on Linear Attention, that have substantial impact despite being direct adaptations of *Katharopoulos et al*’s framework. Highlighted that Linear Attention’s framework itself can be viewed as incremental and that DANet architecture, on the contrary, does not fit into this framework. Enumerated 10 novel and original contributions in our work: numerical stability analysis, MaxNormActivation, $N^{-1/3}$ scaling, complete softmax elimination, reduction in number of heads, altered block structure, projection matrix fusions, Global-Local-ShiftedLocal pattern, industry-competitive 340M encoder pre-training, hardware-agnostic linear-time design, etc, which constitute substantial innovation both individually and in combined synergy.
>
>
> * **W3, Q3 (Baselines)**. Pointed out that modern baselines were already provided in Table 13 which contains 25+ recent Transformer-based methods including Hedgehog (2024) mentioned by the reviewer. Provided additional new results, as requested by the reviewer, showing that DANet outperforms Mamba and RWKV on LRA. Emphasized that, beyond LRA, in real-world LM pre-training (Table 4), DANet surpasses the strongest recent Transformer (MosaicBERT) and bidirectional SSM (BiGS) in the 350-450M size,  <1T-token training settings.
>
> * **Q4 (CLM vs MLM)**. Reaffirmed deliberate focus on bidirectional modeling for maximum real-world impact at limited compute, with plan to open-source our pre-trained competitive sub-quadratic encoder for the community’s benefit. Clarified that CLM results were only intended to demonstrate viability and offered to move Table 5 to the appendix to avoid focus on CLM. Provided detailed exposition of efficient linear-time causal DenseAttention (section **Chunkwise-parallel algorithm**) and committed to adding this exposition to the appendix.
>
> * **Minor Q1, Q2 (Linear Attention and SSMs)**. Discussed conceptual differences of DANet from Linear Attention’s (LA) framework: non-negative feature maps and reweighting are its mandatory parts *by definition*, and they are essential for maintaining stability. Provided citations from the literature (“Rethinking Attention with Performers”, “The Devil in Linear Transformer”), confirming the necessity of these elements for LA. Reiterated that DenseAttention completely eliminates them. Explained that learnable/data-dependent decay/gating is an essential part of SSMs, and fixing it to identity violates the core idea behind this element, makes SSMs numerically unstable and leads to degraded modeling quality, as discussed and empirically confirmed in the relevant literature.
>
> * **Minor Q3 (Native bidirectionality)**. Clarified that "natively bidirectional" means one sequence pass where each query interacts with all keys, while SSMs require two passes (~2x FLOPs) in opposite directions, even with chunkwise-parallel computation. Stressed that although one-pass SSM calculation is mathematically feasible, it’s impractical due to ensuing quadratic complexity.
>
> * **Minor Q4 and Grammar / Rewording / Formatting**. Committed to all requested changes.

---

### Official Review · Reviewer_3qa3 · 2025-11-01

**Soundness:** 2
**Presentation:** 3
**Contribution:** 3
**Rating:** 4
**Confidence:** 3

**Summary:**

This paper proposes DenseAttention, which serves as a drop-in replacement for the classic attention in Transformers. It remove softmax and using MaxNormActivation to ensure numerical stability. DenseAttenion can be computed in O(N^2d) or O(Nd^2) complexity with pure matrix multiplications. Experiments show it achieves competitive performance on long range arena and GLUE tasks while having high throughputacross different GPUs and CPU.

**Strengths:**

1. DenseAttention is well supported in this paper with both mathematical grounding from proposition 1 and 2, and experiments across appropriate tasks such as LRA, GLUE and CLM.
2. DenseAttention achieves impressive hardware efficiency improvements compared with standard and linear attention.
3. Once open sourced,  the lain PyTorch implementation will be a good contribution to community for people to just plug in and use in most hardwares.

**Weaknesses:**

1. While GLUE is a proper task to validate DenseAttention capabilities, there is no modern language model evaluation benchmarks such as MMLU and other reasoning tasks.
2. The scaling effect of DenseAttention is unclear as the model size is around 360M.
3. There is no ablation analysis on the individual components of the DenseAttention, e.g., contribution of removing softmax, using MaxNorm, removing W_k, W_v, W_o, etc.

**Questions:**

1. What do attention patterns that learned by DenseAttention look like and how they are different from softmax attention?
2. How does DenseAttention compare with Linear Attention in detail?
3. Does DenseAttention still main good properties of softmax attention such like long range dependencies and in-context modeling?

---

> ### Author Response · Authors · 2025-11-24
> **Author Response to Reviewer 3qa3. Part 1**
>
> Dear Reviewer 3qa3,
>
> We thank you for your insightful comments and valuable feedback. We sincerely appreciate recognition of the paper's mathematical grounding, the impressive hardware efficiency of our method, and its potential as a practical contribution to the community. Your feedback helped enhance our work as it prompted us to add a new series of ablations. In the following, we hope to address your comments with detailed responses:
>
> ---
>
>
> ### W1. “While GLUE is a proper task to validate DenseAttention capabilities, there is no modern language model evaluation benchmarks such as MMLU and other reasoning tasks.”
>
> Due to limited computational resources which preclude large-scale training runs, we focus primarily on *bidirectional* modeling in this work. It allows for a tangible real-world impact at our parameter scale of 340M, as the original Transformer BERT is still the most downloaded model on HuggingFace. GLUE is the principal benchmark suite for bidirectional language models. And, as we demonstrated in **Table 4**, bidirectional DANet-BERT outperforms key recent industry-competitive baselines among those pre-trained on <1T tokens.
>
> We are actually planning to open-source the weights of this model upon acceptance of our work. We hope it will benefit both academic and industrial communities, as it currently scores highest on GLUE among sub-quadratic bidirectional language models that we are aware of.
>
> On the other hand, MMLU and, even more so, reasoning benchmarks are dedicated to evaluating exclusively *autoregressive* Large Language Models rather than bidirectional moderate-size ones. To produce statistically significant results above chance on such benchmarks, a model needs to have size at least in the low billions of parameters and to be pre-trained on hundreds of billions to trillions of tokens. This is exemplified in Table 4 (page 38) of this recent survey: https://arxiv.org/abs/2411.03350v2 where all models below 3B parameters score uniformly near random (25%) on MMLU despite vastly different sizes and up to hundreds of billions tokens pre-training. Notable exceptions are 1-3B Llama 3.2 and Gemma 2 which are distilled from larger models trained on 15T and 13T respectively. The same considerations are applicable to other commonsense reasoning benchmarks (ARC, Winogrande) for sub-0.5B models.
>
> Since the autoregressive DANet was pre-trained on 11B and it has 360M parameters, it wouldn't produce meaningful results on these benchmarks. However we assured that it is competitive with Transformer via other metrics (perplexity) in **Table 5**. We emphasize again that causal language modeling is auxiliary in our work as, having limited computational resources, we prioritized making real-world impact and deployed the bulk of the resources to pre-train a competitive encoder model.
>
> ---
>
>
> ### W2. “The scaling effect of DenseAttention is unclear as the model size is around 360M.”
>
> We thank you for voicing the concern about scaling properties of the architecture. A scaling study we had conducted (**Appendix B.4**, specifically Table 11 on lines 1344-1349 in the manuscript) indicates that DenseAttention architecture indeed exhibits favorable scaling properties similar to standard Transformer, as modeling quality grows with the parameters count. We argue that, given this evidence, the trend should continue as the model size gets extended to the billions of parameters, fully analogous to Transformer scaling.
>
> If your concern is rather about the maximal model size used in experiments, please let us offer a perspective on this matter. Firstly, this size appears to be fairly standard and most popular for industrial applications to this day for the encoder models (as mentioned earlier about the most downloaded model on HF). Secondly, many of the works on novel architectures that have been recently published in leading ML venues by reputable author collectives and well received by the community (judging by their citation counts and distinctions by conference committees), employ similar-sized or smaller models for language modeling validation. Here are some examples:
>
> * S4 (ICLR Oral, 2022) – 250M parameters. https://openreview.net/pdf?id=uYLFoz1vlAC
>
> * Toeplitz Neural Network (ICLR notable top 25%, 2023) – 126M parameters. https://openreview.net/pdf?id=IxmWsm4xrua
>
> * Monarch Mixer (NeurIPS Oral, 2023) – 360M parameters. https://openreview.net/pdf?id=cB0BImqSS9
>
> * The Hedgehog & the Porcupine (ICLR, 2024) – 125M parameters. https://openreview.net/forum?id=4g02l2N2Nx
>
> * Elliptical Attention (NeurIPS, 2024) – 90M parameters for dense models. https://openreview.net/forum?id=Ejg4d4FVrs
>
>
> These examples and the peer researcher’s appreciation for them imply that a 360M parameters model scale is not generally perceived as a limitation or weakness by the community.
>
> ---

---

> > ### Author Response · Authors · 2025-11-24
> > **Author Response to Reviewer 3qa3. Part 3**
> >
> > However, we notice an inefficiency in that configuration: as discussed in Section 3.1, projection matrices $W_q$ and $W_k$ become mutually redundant, as well as $W_v$ and $W_o$. To mitigate this redundancy, we keep only parameters $W_q$ and $W_v$, effectively fusing them with $W_k$ and $W_o$, respectively. But this modification reduces the parameter count to 5/6 of that in a standard Transformer layer. To compensate, we increase the number of layers from 24 to 29. The new variant $(6 \Longrightarrow 7)$ robustly surpasses the speed of FlashAttention-augmented Transformer for the first time in the chain of ablations, even despite having a greater number of layers.
> >
> > Yet, there is still room for improvement in efficiency: as detailed in Appendix C, residual connections and layer normalizations are memory-bound, computationally inefficient operations, and removing them from between the attention and FFN sub-layers brings an additional benefit of redundancy of the $W_v$ and first FFN projection matrices, allowing for their fusion. Accordingly, we remove the residual connection between attention and FFN, move MaxNorm from there to immediately after the FFN, and merge the two matrices into one. This final modification $(7 \Longrightarrow 8)$ produces DANet.  As the parameter count in a DANet layer is yet lesser, than in $(7)$, we increase the number of layers to 32. Finally, we observe that DANet delivers the highest throughout in the chain of ablations in Table A, outperforming FlashAttention by 6%.
> >
> > ---
> >
> > ### Questions
> >
> > > Q1. “What do attention patterns that learned by DenseAttention look like and how they are different from softmax attention?”
> >
> > Thank you for the insightful question! DenseAttention’s patterns are essentially very similar to `1+ELU` and `ReLU` cases in Figure 7 of “The Hedgehog & the Porcupine” paper you mentioned earlier (https://openreview.net/forum?id=4g02l2N2Nx). They are more dispersedly and evenly distributed across the attention scores matrix with slight concentration around the main diagonal ($i=j$) while softmax attention tends to concentrate the probability weight there to a much greater degree.
> >
> > > Q2. “How does DenseAttention compare with Linear Attention in detail?”
> >
> > We genuinely thank you for your interest. We tried to comprehensively compare DenseAttention with Linear Attention and derivative architectures from a **theoretical standpoint** in Appendix **Related Work: Sub-quadratic Algorithms for Sequence Processing** (start at line 1134), as well as **empirically** in language modeling performance (*Table 4*), general sequence processing across different modalities (*Table 13* starting on line 1513) and in speed (**Table 6**). If you have any specific questions on this topic remaining, we will be happy to elaborate on them, too.
> >
> > > Q3. “Does DenseAttention still main good properties of softmax attention such like long range dependencies and in-context modeling?”
> >
> > Yes. It is validated by strong performance on Long Range Arena (tasks with up to 16K context size, including language modeling and logical reasoning) as shown in **Table 1** and **Table 13**, by ablations in language modeling on long contexts (up to 16K) presented in **Table 10**, and by the overall best result on the Pathfinder-256 task (65K context size) in **Table 9**.

---

> ### Author Response · Authors · 2025-11-24
> **Author Response to Reviewer 3qa3. Part 2**
>
> ### W3. “There is no ablation analysis on the individual components of the DenseAttention.”
>
> **Prior  ablations**. Thank you for this insightful comment. First, we’d like to draw your attention to existing ablations, presented in the paper. We had to defer many of them to the appendix due to the space limit of 9 pages. We are sorry for any confusion it might have caused and list them here for your convenience: 1) number of heads in **Table 3** motivating fused-matrix block structure; 2) LayerNorm vs MaxNorm in **Table 12**; 3) Positional encodings in **Table 2 and Table 14**; 4) Local–ShiftedLocal–Global attention layers pattern in **Table 2 and Table 10**.
>
> Please let us also clarify specific points raised in your comment:
>
> “contribution of removing softmax” – this modification cannot be examined on its own, separate from other architectural innovations in our work. When we remove softmax from Transformer, it simply *fails to learn*, producing *NaNs* and *infs* shortly after the training onset. We communicated this on lines 193-195 (we will change the phrasing from “can lead” to “leads” for extra clarity).
>
> “using MaxNorm” – we provided the ablation on use of MaxNorm in the Appendix **B.5. Analysis of MaxNorm and Layer Norm**, specifically in the **Table 12** (lines 1393-1402).
>
> “removing W_k, W_v, W_o” – this is a direct consequence of reduction in the number of heads, as we discussed in lines 260-269 in the manuscript. It makes these matrices redundant and necessitates their merging with other matrices in order not to waste computation. We presented the ablation on the reduction of the number of heads in the main body of the paper, in **Table 3**.
>
> **New ablations**. Motivated by your feedback, we resolved to conduct additional ablations. We again thank you for this useful suggestion and present them below.
>
> ---
>
> **From Transformer to DANet**
>
> We analyze differences between Transformer and DANet architectures through the lens of efficiency and stability, following individual changes in design step-by-step and measuring inference throughput at sequence length 512 and model size of 340M parameters. In this ablation, all models except $(1a)$ are implemented in plain Pytorch. All measurements were conducted on a Nvidia A100 80GB SXM in fp16 numerical precision.
>
>
> | #  | Model description                                                | Stability / Inference Throughput |
> |----|------------------------------------------------------------------|----------------------------------|
> | 1  | Transformer, 16h                                                 | 112.92                           |
> | 1a | Transformer (FA implementation), 16h                             | 277.66                           |
> | 2  | No softmax, 16h                                                  | FAILS                            |
> | 3  | No softmax, scaling by $N^{-1/3}$, 16h                         | 273.94                           |
> | 4  | No softmax, scaling by $N^{-1/3}$, 4h                          | FAILS                            |
> | 5  | No softmax, scaling by $N^{-1/3}$, MaxNorms, 4h                | 272.18                           |
> | 6  | No softmax, scaling by $N^{-1/3}$, MaxNorms, 1h                | 277.17                           |
> | 7  | No softmax, scaling by $N^{-1/3}$, MaxNorms, no $W_k$, $W_o$, 1h | 284.77                     |
> | 8  | DANet, 1h                                                        | 294.11                           |
>
> Table A. Inference throughput, thousands of tokens per second of different architectural modifications, leading from Transformer to DANet.
>
>
> Removing softmax from standard Transformer with 340M parameters, 24 layers and 16 heads leads  $(1 \Longrightarrow 2)$ to numerical instabilities and eventual training divergence after just 1 billion training tokens. Introduction of a scaling factor of $N^{-1/3}$ between layer normalization and the attention layer $(2 \Longrightarrow 3)$ recovers stability, however, reducing number of heads from 16 to 4 $(3 \Longrightarrow 4)$ again results in numerical instability and failure to learn. The ultimate stability is achieved only when standard LayerNorms get replaced by MaxNorms $(4 \Longrightarrow 5)$. We observe that variants $(3)$ and $(5)$ are already 2.4 times faster than standard PyTorch Transformer, and are almost as fast as FlashAttention low-level implementation. Further reducing the number of heads to 1 $(5 \Longrightarrow 6)$ bridges the gap with FlashAttention in agreement with the insights from Appendix D.

---

> ### Author Response · Authors · 2025-12-03
> **Summary of Rebuttal for Reviewer 3qa3**
>
> * **W1**: We emphasized the focus at bidirectional modeling in our work, which has enabled tangible real-world impact (best encoder LM of similar size and training duration on GLUE). Reaffirmed the appropriateness of the GLUE benchmark for this modality, in contrast to MMLU and others suggested by the reviewer. Discussed suitable architectures (autoregressive) and appropriate model scales (>2-3B) for the suggested benchmarks to run and make sense, rendering them out of scope for current research. Provided extended evidence that empirically supports these facts. Referred to meaningful CLM validation results included in our paper.
>
>
> * **W2**: Referred to the results from the scaling study in the paper indicating that DANet exhibits favorable scaling properties, similar to Transformer. Communicated that 340M params is standard and most popular scale for encoder architectures in industry to this date. Highlighted 5 highly regarded papers, similar in scope to ours and published in ICLR and NeurIPS (including 3 orals). Noted that all of them use maximum model sizes less than or equal to our scale which reflects the community’s acceptance and appreciation of such validation scales.
>
> * **W3**: Pointed to 4 sets of ablations already provided across 6 tables in the manuscript and made further clarifications about specific points in the reviewer’s comment (“removing softmax, using MaxNorm, removing W_k, W_v, W_o”). Presented a **new series of ablations** and their analysis (**From Transformer to DANet**) which isolates individual modifications, as requested by the reviewer.
>
> * **Q1**: Answered that the patterns are similar to `1+ELU` and `ReLU` cases in “The Hedgehog & the Porcupine” paper.
>
> * **Q2**: Referred to relevant parts of the paper which answer the question in detail both theoretically and empirically.
>
> * **Q3**: Affirmed the premise and pointed to 4 tables in the paper that support it.

---

### Author Response · Authors · 2025-12-04
**General Summary**

We sincerely thank reviewers **3qa3**, **RdSr**, and **yhrB** for their time, authentic feedback, and constructive suggestions which led to additional experiments and helped to clarify and strengthen our work. We would like to emphasize the highlighted main points and strengths of our paper, and key updates prompted by the feedback.

---

We are grateful to reviewers for acknowledging the following details and strengths in our work:

* The paper proposing **DenseAttention Network (DANet)** as a **simplified**, **drop-in replacement** for classic Transformer attention, achieving **sub-quadratic complexity** in runtime and memory for fast processing of long sequences while retaining a **simple architecture** close to Transformers to facilitate adoption (3qa3, RdSr, yhrB), with **conceptual minimalism** in reducing attention to **weight-shared matmuls** that is **novel** and may **inspire further work** in simplified Transformer desings (yhrB).

* The architecture being **well-justified** and **carefully motivated**, including **mathematical grounding** from propositions, **ablation studies** strengthening the analysis, and a **solid overall story** with clear presentation of main goals (3qa3, RdSr).

* Impressive **hardware efficiency improvements** and **much higher throughput** across sequence lengths and hardware compared to standard Transformers and linear attention alternatives, with **consistent advantages** on CPUs, older GPUs, and other devices without relying on specialized kernels or device-specific optimizations (3qa3, RdSr, yhrB).

* Strong empirical results showing DANet achieves **competitive performance** and **rivals and beats** similarly-sized Transformer-based and sub-quadratic models on language modeling tasks like **GLUE**, **Long Range Arena**, and **CLM** (3qa3, RdSr).

* The approach being **device-agnostic** and **efficient**, addressing a **pressing challenge of efficient attention** for scaling language models with a **simple and broadly applicable design** (RdSr, yhrB), offering **potential for impact** and influence as a general Transformer improvement (yhrB).

* The **plain PyTorch implementation** serving as a **good contribution** to the community for easy plug-in use across most hardwares (3qa3).

---


Here's the summary of the key updates and responses we provided based on reviewers’ comments and suggestions.

* Conducted a comprehensive step-by-step ablation from Transformer to DANet (detailed in **Table A/C** of the rebuttals to **Reviewer 3qa3/ yhrB**, to be added as new **Appendix B.6** in the final revision), which demonstrates how each individual modification contributes to numerical stability and superior inference throughput.

* Presented an additional ablation studying depth-vs-width trade-off (24 layers with doubled hidden dimension d=2048 vs standard 32 layers, d=1024) in **Table D** of the rebuttal to **Reviewer yhrB** (to be added as **Appendix B.7**) , showing that both approaches yield comparable strong performance while multi-head variants become beneficial at larger hidden sizes.

* Provided results that DANet surpasses additional SSM/ RNN baselines (Mamba and RWKV) on the LRA benchmark suite (RdSr). Performed an experiment on **hybrid DenseAttention + local softmax attention architecture** (**Table A** of rebuttal to **Reviewer yhrB**), which outperformed both linear and quadratic MEGA variants. Additionally conducted experiments with augmented data setup from “On the Locality Bias…”, demonstrating DANet's competitive performance in this setting (**Table B** of rebuttal to **Reviewer yhrB**).

* Provided **comprehensive justification for 340-360M parameter scale** in rebuttals to **Reviewers 3qa3 and RdSr** with examples from recent influential works (S4, Monarch Mixer, Toeplitz, Hedgehog & Porcupine, Elliptical Attention) demonstrating community acceptance of similar or smaller scales for architectural validation, and emphasizing practical impact for bidirectional models at this scale.

* Provided detailed exposition of the **chunkwise-parallel algorithm** for causal attention adapted to DenseAttention in the rebuttal to **Reviewer RdSr** (to be added as new **Appendix F** in final revision), explaining how it efficiently computes causal attention in linear time through chunk-based parallelization.

* Committed to incorporate citations, clarifications and fixes of grammar/ formatting into the final revision as suggested by **Reviewers RdSr and yhrB**.

---

### Meta-Review · Area_Chair_JFtE · 2025-12-08

**Summary:**

This paper proposes a novel architecture to replace attention, that scales linearly with sequence length.

I have discarded the review of rev.cbXF which is very likely LLM-generated and does not contain novel insights compared to the other reviews.

The reviewers agree about several of the paper's strengths:
- plain, device agnostic linear architecture
- theoretical discussion about removing softmax and $l_\infty$ normalization is interesting and non-trivial
- throughputs improvements at a fixed 340M scale, especially impressive at larger sequence length

These strengths make this research interesting and worth pursuing.

However, the reviewers also raised concerns regarding the possible lack of scaling of the method:
- Showing improvements on downstream tasks is interesting, but it would be worthwhile to also show validation loss curves, as this metric is often much less noisy.
- The paper proposes an architectural change, which changes the throughput and number of parameters of the network in a non-trivial way. It would therefore be better to quantify improvements in a (budget, performance) 2-d pareto front, when the number of parameters of DANet and of a standard transformer are changed; here it would be great to look at budget both at inference and at training time. The rebuttal adds insightful ablations but a clear picture is still missing.

- Since lack of compute budget is a problem for the authors, it would be worthwhile to indicate scaling behavior in a more detailed way than table 11. In order to get an insight of the scaling behavior, it would be beneficial to add scaling laws to the paper: train several small scale models of different sizes (e.g. 50, 100, 200, 400M) parameters, and then fit a Chinchilla style scaling law L = E + A/N^\alpha + B / D^\beta for both DANet and a standard transformer; which would tell how the proposed architecture scales vs a transformer.

**Reviewer Concerns:**

##  Lack of demonstration of scalability (3qa3, RdSr, yhrB)
This concern is still outstanding, cf summary above

## Ablation on the different components introduced here ( 3qa3,  yhrB)

The authors convincingly isolate the effect of each component in the rebuttal.

## Other evaluations (3qa3)

Since the scale of the present paper is quite small, I agree with the authors that adding other more complex evaluations is too ambitious. However, I would argue that a more in-depth study of validation losses would greatly improve the paper.


## Comparison to new baselines / more in-depth comparions (3qa3,  RdSr,   yhrB)

The authors condignly clarify the scope of the paper in the rebuttal.

## Causal transformers (RdSr)

Even though I agree with the reviewer that expanding the paper in the direction of causal transformers would strengthen it greatly, I think that the paper is clear enough that the focus is on bert style architecture, which are widely used an useful.

**Reviewer Scores:**

3qa3 might have increased their score to a 6, since their concerns are mostly alleviated.

RdSr might have bumped their score to a 4, seeing how some of their concerns are maintained.

Similarly, rev. yhrB might have increased their score to a 4.

This, however, depends on how much the reviewers are concerned by the "scaling" of the algorithm; I feel like the authors could address that question more clearly. If this point is as critical for the reviewers as it is for me, I think their score would have stayed the same.

---

### Decision · Program_Chairs · 2026-01-26

Reject